# TRAINING BINARY NEURAL NETWORKS IN A BINARY WEIGHT SPACE

## ABSTRACT

Binary neural networks (BNNs), which have binary weights and activations, hold significant potential for enabling neural computations on low-end edge devices with limited computational power and memory resources. Currently, most existing BNN training approaches optimize and binarize real-valued weights, leading to substantial memory usage during training. Though training BNNs without real-valued weights to save memory is intriguing, it has been deemed challenging with gradient-based optimization. To address this challenge, we define an update probability for binary weights, determined by the current binary weights and real-valued gradients. The binary weights generated by our method match those obtained by SGD in the real-space training of BNNs in the expectation. As a result, the training of binary weights becomes stable even without real weights. Our method yields image classification performance comparable to baselines that utilize real weights, yet reduce memory usage by up to a factor of 33.

## 1 INTRODUCTION

Neural networks (NNs) have achieved significant breakthroughs in various research fields such as image recognition and natural language processing (Bahdanau et al., 2015; He et al., 2016a; Dosovitskiy et al., 2021). The demand for NNs, therefore, extends to resource-constrained edge devices including industrial quality control terminals, domestic gadgets, and wearable devices. Consequently, there has been an increasing number of studies in reducing NN memory usage during *inference*. Knowledge distillation (Hinton et al., 2015; Yim et al., 2017), neural architecture search (Zoph & Le, 2017; Mellor et al., 2021), network pruning (Li et al., 2017; Meng et al., 2020), and tensor decomposition (Li et al., 2009; Anandkumar et al., 2014) aim to construct compact and high-performing models. Network quantization methods replace high-precision trained weights with low-precision ones in inference, such as 4-bit, 2-bit, or even 1-bit (Bhalgat et al., 2020; Esser et al., 2020; Gupta et al., 2015; Liu et al., 2021b), where the 1-bit case is referred to as binary neural networks (BNNs).

BNNs (Hubara et al., 2016), having binary weights and activations, offer significant memory savings compared to real NNs. The multiply-add operations in BNNs can be implemented using hardware-friendly XNOR and bit-count operations, eliminating the need for computationally expensive floating-point operations or fixed-point overflow handling. Rastegari et al. (2016) demonstrated a 58-times speed-up in inference with BNNs, making real-time inference feasible on edge devices.

Unlike inference, training BNNs requires nontrivial techniques. Specifically, when certain layers employ a piecewise constant activation function like the sign function, all the layer-wise gradients for the loss function up to that layer vanish. As a result, direct application of gradient descent is infeasible. Among methods tackling this issue (Liu et al., 2018; Gong et al., 2019), the straight-through estimator (STE) (Bengio et al., 2013) replaces the first derivative of the activation function with a piecewise

Table 1: Weight precision [bit]. FP indicates the floating-point format, *i.e.*, typically 32-bit. Our goal is to train BNNs without retaining real-valued weights, which is different from the traditional BNNs training setup.

|  | Training | Inference |
|---|---|---|
| LSQ (Wang et al., 2021) | FP | $\geq 2$ |
| Naive STE (Hubara et al., 2016) | FP | 1 |
| ScaleGrad (Sun et al., 2020) | $\geq 4$ | $\geq 4$ |
| MIP (Kurtz & Bah, 2021) | 1 | 1 |
| Ours (binary-space training) | **1** | **1** |

constant function to prevent gradient vanishing, and recent frequency domain approximation (Xu et al., 2021) approximates the sign function in the Fourier domain to obtain a pseudo-gradient that preserves the genuine gradient direction.

Although these methods have improved BNNs' performance, they require maintaining real-valued weights alongside the binary weights during *training*, which consumes a significant amount of memory. Reducing memory consumption during training is crucial for enabling on-edge deep learning, which is in high demand; for example, an edge AI system for crop quality control on farms without WiFi access may need to be locally updated, or a home monitoring system may require personalization without sending private information outside. Several studies have focused on reducing memory use during training as summarized in Table 1. Wang et al. (2021) proposed a gradient-quantization method to train BNNs, demonstrating a 3.7 times reduction in memory compared to conventional BNNs while retaining real-valued weights throughout training. Another study attempted to train binary weights directly by using fully model-based discrete optimization methods (Kurtz & Bah, 2021), including mixed-integer programming (MIP). However, these methods have been evaluated only on datasets smaller than MNIST due to significant training-time constraints. Sun et al. (2020) introduced a method to train 4-bit weights by quantizing gradients on a log scale and applying stochastic gradient descent (SGD). However, it is commonly believed that the use of SGD requires maintaining a certain degree of precision in the weights as accumulators (Hubara et al., 2016), and it is challenging to extend SGD directly to training 1-bit weights since encoding gradient direction alone consumes 1-bit. While binary (1-bit) weights are computationally efficient, to the best of our knowledge, none of the previous methods have successfully trained BNNs on practical datasets such as CIFAR-100 without the use of real-valued weights.

To mitigate memory consumption during training, we introduce a novel BNN training algorithm that eliminates the need for real-valued weights. Our method calculates an appropriate probability for binary weights to flip their sign so that the loss is expected to decrease. To derive this probability, we model the distribution of real-valued weights and design a stochastic masking method that reflects the magnitude of the real-valued gradients to emulate weight updates in real-space training.

Our contribution is three-fold: 1) We have introduced a memory-efficient BNN training method that does not require the retention of real weights. We define an update probability for binary weights as a function of the current binary weights and approximations of real gradients. The binary weights produced by our update rule match those obtained by the conventional BNN training method that uses real weights in expectation. 2) We have successfully trained BNNs using the proposed algorithm on relatively large datasets, such as CIFAR-100, for the first time in binary weights-only training. The performance of our method is comparable to that of an existing method retaining real weights. 3) We have analytically evaluated the memory footprints required by BNN training methods and demonstrated that our method can reduce memory usage by a factor of 33 compared to the conventional method retaining real weights in scenarios with infinite hidden layers and units.

## 2 RELATED WORK

A binary neural network (BNN) (Courbariaux et al., 2015; Hubara et al., 2016) is a variant of NNs that utilizes binary weights and propagates binary activation values to achieve efficient inference. Numerous efforts have been dedicated to developing architectures (Rastegari et al., 2016; Darabi et al., 2018; Bulat & Tzimiropoulos, 2019; Martinez et al., 2020; Qin et al., 2020; 2023) and gradient approximation methods (Gong et al., 2019; Cai et al., 2017) for BNNs. Through these studies, a learning paradigm that optimizes *real-valued* weights and binarizes them using a sign function during propagation has emerged as one of the most successful approaches for obtaining trained binary weights. During training, BNNs propagate binary activation values at each layer $l = 1, \cdots, L$ as

$$\boldsymbol{h}_l = \mathrm{sgn}(\boldsymbol{a}_l), \ \ \boldsymbol{a}_l = \mathrm{sgn}(\boldsymbol{\omega}_l)\boldsymbol{h}_{l-1}, \ \ \text{where} \ \ \mathrm{sgn}(x) = \begin{cases} 1 & (x \geq 0) \\ -1 & (x < 0) \end{cases}, \tag{1}$$

where $\boldsymbol{h}_l$ is an output, $\boldsymbol{\omega}_l \in \mathbb{R}^{D_l \times D_{l-1}}$ is a real-valued weight matrix, $D_l$ is a dimension of $l$-th layer and $L$ is the number of layers. To compute the pseudo-gradient of the loss function, the STE (Bengio et al., 2013) is widely adopted to address the gradient vanishing issues caused by the sign activation function; it approximates the derivative of the sign function by that of a *hard tanh* function (Hubara et al., 2016) during backward propagation. Given a loss function $\mathcal{L}$, it computes a pseudo-gradient of

**Algorithm 1:** Training BNN in binary space

**Requires:** *Mini-batch* $(X, Y)$*, MaskGenerator*
Forward process:
$\boldsymbol{h}_0 = X$
**for** $l = 1$ **to** $L$ **do**
$\quad \boldsymbol{a}_l = \text{BatchNorm}_{0,1}(\boldsymbol{w}_l \boldsymbol{h}_{l-1})$
$\quad \boldsymbol{h}_l = \text{sgn}(\boldsymbol{a}_l)$
Backward process:
$\mathcal{L} = \text{SoftmaxCE}(Y, \boldsymbol{a}_L)$
**for** $l = L$ **to** $1$ **do**
$\quad \boldsymbol{g}_l \leftarrow \nabla_{\boldsymbol{w}_l} \mathcal{L}, \; \boldsymbol{w}_l^* \leftarrow [\![-\boldsymbol{g}_l \geq 0]\!]$
$\quad \boldsymbol{m}_l \leftarrow \text{MaskGenerator}(\boldsymbol{w}_l, \boldsymbol{g}_l)$
$\quad \boldsymbol{w}_l \leftarrow \overline{\boldsymbol{m}}_l \cdot \boldsymbol{w}_l + \boldsymbol{m}_l \cdot \boldsymbol{w}_l^*$

**Algorithm 2:** Training BNN in real space

**Requires:** *Mini-batch* $(X, Y)$*, Learning rate* $\eta$
Forward process:
$\boldsymbol{h}_0 = X$
**for** $l = 1$ **to** $L$ **do**
$\quad \boldsymbol{a_l} = \text{BatchNorm}_{\mu_l, \sigma_l}(\text{sgn}(\boldsymbol{\omega}_l)\boldsymbol{h}_{l-1})$
$\quad \boldsymbol{h}_l = \text{sgn}(\boldsymbol{a}_l)$
Backward process:
$\mathcal{L} = \text{SoftmaxCE}(Y, \boldsymbol{a}_L)$
**for** $l = L$ **to** $1$ **do**
$\quad \boldsymbol{g}_l \leftarrow \nabla_{\text{sgn}(\boldsymbol{\omega}_l)} \mathcal{L}, \; \boldsymbol{\omega}_l^* \leftarrow \boldsymbol{\omega}_l - \boldsymbol{g}_l$
$\quad \boldsymbol{\omega}_l \leftarrow (1 - \eta)\boldsymbol{\omega}_l + \eta \boldsymbol{\omega}_l^*$

$\mathcal{L}$ with respect to $\boldsymbol{a}_l$ and $\boldsymbol{\omega}_t$ by

$$\frac{\partial \mathcal{L}}{\partial \boldsymbol{a}_l} \approx \frac{\partial \mathcal{L}}{\partial \boldsymbol{h}_l} \circ \mathbf{1}_{|\boldsymbol{a}_l| \leq 1}, \quad \frac{\partial \mathcal{L}}{\partial \boldsymbol{\omega}_l} \approx \frac{\partial \mathcal{L}}{\partial \text{sgn}(\boldsymbol{\omega}_l)}. \tag{2}$$

These gradient approximations enable the training of real-valued weights using stochastic gradient descent (SGD) methods. While STE has been successful in training BNNs, it typically requires real-valued weights, which implies substantial memory usage during *training*.

## 3 TRAINING BNNs IN A BINARY WEIGHT SPACE

Since binary weights have extremely low precision, direct optimization in the binary weight space based on gradients is not straightforward. In this study, we attempt to address this challenge by extending SGD to the binary space and deriving a stochastic update rule.

### 3.1 ALGORITHM

**Overview and notation.** Let $\boldsymbol{w} \in B^N$ be a binary vector of flattened learnable parameters of a certain layer of a BNN. While many previous studies use $B = \{1, -1\}$, we use $B = \{0, 1\}$ to make the discussion concise with Boolean operations: OR $a + b$, AND $a \cdot b$, and NOT $\overline{a}$ for $a, b \in B$. With time index $t = 0, 1, \cdots$, we denote by $\boldsymbol{w}_t$ the binary weights at time $t$ and assume that $\boldsymbol{w}_0$ is randomly initialized. The overview of our binary-space training is shown in Algorithm 1. Inspired by SGD, our algorithm iteratively updates the binary weights. The main flow of the algorithm at time $t$ takes the following four steps: (1) Given a loss function $\mathcal{L}$, the real gradient $\boldsymbol{g}_t = \nabla_{\boldsymbol{w}_t} \mathcal{L} \in \mathbb{R}^N$ is computed. (2) We compute $\boldsymbol{w}_t^* = [\![-\boldsymbol{g}_t \geq 0]\!] \in B^N$, which we call target weight[1], where $[\![x]\!]$ takes 1 if $x$ is True and 0 otherwise. (3) A *hypermask* $\boldsymbol{m}_t \in B^N$ is sampled from a conditional probability distribution over $B^N$ given $\boldsymbol{g}_t$ and $\boldsymbol{w}_{t-1}$. (4) The binary weights are updated as follows:

$$\boldsymbol{w}_t = \overline{\boldsymbol{m}}_t \cdot \boldsymbol{w}_{t-1} + \boldsymbol{m}_t \cdot \boldsymbol{w}_t^*, \tag{3}$$

where Boolean operations are applied element-wise.

**Analogy to SGD in real space.** The update rule in Eq (3) is motivated by SGD in Algorithm 2. With SGD, the following update rule is applied to train a real-valued weight $\boldsymbol{\omega}_t \in \mathbb{R}^N$

$$\boldsymbol{\omega}_t = (1 - \eta)\boldsymbol{\omega}_{t-1} + \eta \boldsymbol{\omega}_t^*, \tag{4}$$

where $\eta$ is a hyperparameter (learning rate) and $\boldsymbol{\omega}_t^* = \boldsymbol{\omega}_{t-1} - \boldsymbol{g}_t$ is what we call target weight. When $\eta \in [0, 1]$, this update rule is equivalent to replacing the point $\boldsymbol{\omega}_{t-1}$ by a certain point $\boldsymbol{\omega}_t$ that satisfies

$$\boldsymbol{\omega}_t \in \underset{\boldsymbol{p} \in \mathbb{R}^N}{\arg\min} \left( \|\boldsymbol{\omega}_{t-1} - \boldsymbol{p}\|_2 + \|\boldsymbol{\omega}_t^* - \boldsymbol{p}\|_2 \right) \tag{5}$$

because $\boldsymbol{\omega}_t$ is an interpolation of $\boldsymbol{\omega}_{t-1}$ and $\boldsymbol{\omega}_t^*$ as shown in Figure 1a. Analogously, binary weight update in the binary metric space $(B^N, d)$ can be chosen from a set of candidate points given by

$$\boldsymbol{w}_t \in C_t := \underset{\boldsymbol{p} \in B^N}{\arg\min} \left( d(\boldsymbol{w}_{t-1}, \boldsymbol{p}) + d(\boldsymbol{w}_t^*, \boldsymbol{p}) \right), \tag{6}$$

---

[1]We simply call $w_t^*$ target weight as it represents a *direction*, to which a point that reduces the loss is found. It is noted that such a point may not be located in the binary coordinates and $w_t^*$ might even increase the loss.

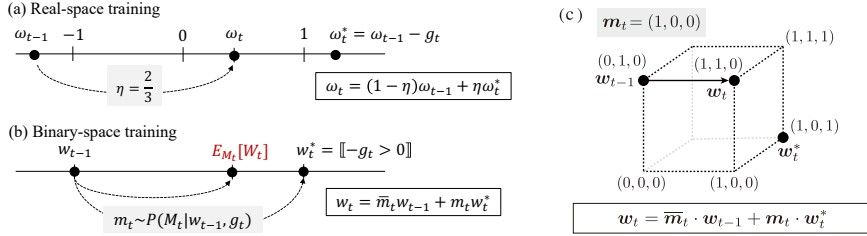

Figure 1: Update rules. (a) In SGD, the magnitude of the update is adjusted by the learning rate and gradient. (b) In binary-space training, a hypermask is used to update binary weights, which controls the expected value of the magnitude of the update. (c) Example of the binary-space update in $B^3$.

where $d(\cdot, \cdot)$ is a distance function in $B^N$. In the case of Hamming distance $d(\boldsymbol{a}, \boldsymbol{b}) = |\{i = 1, \ldots, N : a_i \neq b_i\}|$, the candidate set becomes

$$C_t = \{\boldsymbol{w}_t : \boldsymbol{w}_t = \overline{\boldsymbol{m}}_t \cdot \boldsymbol{w}_{t-1} + \boldsymbol{m}_t \cdot \boldsymbol{w}_t^*, \ \ \boldsymbol{m}_t \in B^N\}. \tag{7}$$

By choosing a particular binary vector $\boldsymbol{m}_t$, which we call hypermask, $\boldsymbol{w}_t$ is uniquely defined as in Eq. (3). In this way, the hypermask in the binary update rule plays an analogous role to the learning rate in the real update rule. As the hypermask governs the learning dynamics in the binary weight space, the design of the high-dimensional hypermask is crucial.

**Intuitive picture.** Figure 1 shows the update rule of Eq. (3) and its analogy to SGD. Suppose we have a three-dimensional weight $\boldsymbol{w}_{t-1} = (0, 1, 0)$ and target weight $\boldsymbol{w}_t^* = (1, 0, 1)$ at time $t$. Then the candidate set $C_t$ is the set of corner points of a hypercube between $\boldsymbol{w}_{t-1}$ and $\boldsymbol{w}_t^*$. By specifying a hypermask $\boldsymbol{m}_t \in B^3$, one of the corner points is chosen as $\boldsymbol{w}_t$.

## 3.2 DESIGN OF HYPERMASKS

In real-space training, a fixed learning rate across multiple mini-batches is still effective as real gradients reflect their magnitudes. However, in binary-space training, the magnitude of update $\boldsymbol{w}_t^* - \boldsymbol{w}$ always takes 0 or 1 and the magnitude of the real gradient cannot be reflected when a *fixed* hypermask is used. Therefore, we explore the stochastic construction of hypermasks.

**Notation and setting** We consider two update rules of binary weights. Given observations of a real-valued weight $\omega_{t-1} \in \mathbb{R}$ and gradient $g_t \in \mathbb{R}$, the first rule, *real-space update*, updates binary weights in a real space in the following two steps: 1) update the real-valued weight as $\omega_t = \omega_{t-1} - \eta g_t$, 2) update the binary weight as $\hat{w}_t = [\![\omega_t \geq 0]\!]$. This is the conventional training rule for BNNs. The second rule, *binary-space update*, updates binary weights in a binary space by using a hypermask $m_t \in B$ as $w_t = \overline{m}_t \cdot w_{t-1} + m_t \cdot [\![-g_t \geq 0]\!]$ where $w_{t-1} = [\![\omega_{t-1} \geq 0]\!] \in B$. Figure 2 compares these update rules. Note that notation is simplified to scalar to improve the readability but extension to multiple dimensions is straightforward. The capital letters of the above variables are used to indicate the random variables. For example, $\Omega_t$ and $W_t$ are random variables of $\omega_t$ and $w_t$, respectively. When there is no confusion, the probability $P(W_t = w_t)$ is simply denoted by $P(w_t)$.

In the following, we introduce two *good* properties to emulate the real-space update by the binary-space update. The first property guarantees that the expected value of the binary-space update will match that of the real-space update, and the second property maximizes the probability that the binary-space update matches the real-space update.

**Definition 1 (Expectation matching property).** *We say that the mask distribution* $P(M_t|w_{t-1}, g_t)$ *satisfies the expectation matching property (EMP) if the following condition is satisfied:*

$$\mathbb{E}_{W_t \sim P(W_t|w_{t-1}, g_t)}[W_t] = \mathbb{E}_{\hat{W}_t \sim P(\hat{W}_t|w_{t-1}, g_t)}[\hat{W}_t]. \tag{8}$$

Intuitively, the magnitude of the update is adjusted in expectation as shown in Figures 1b.

**Definition 2 (Matching maximization property).** *We say that the mask distribution* $P(M_t|w_{t-1}, g_t)$ *satisfies the matching maximization property (MMP) if it maximizes the matching probability MP between* $W_t$ *and* $\hat{W}_t$ *given by*

$$\text{MP} = \sum_{b \in B} P(W_t = b|w_{t-1}, g_t) P(\hat{W}_t = b|w_{t-1}, g_t). \tag{9}$$

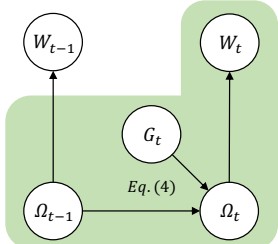 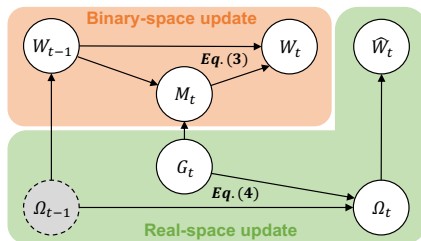

Figure 2: Graphical model for the real- (left) and binary-space training (right, ours). (Left) The real weight $\Omega_t$ is updated after the observation of $\Omega_{t-1}$ and $G_t$. (Right) The binary-space update defines the mask distribution to simulate the real-space update which assumes the distribution of $\Omega_{t-1}$.

Next, we give mask distributions that satisfy the EMP and MMP. To derive them, we first model the probability that the sign of the binary weight flips in the real-space update. In the binary-space training, the real-valued weights do not exist; thus, we assume that $\Omega_{t-1}$ obeys a certain distribution.

**Theorem 1 (Sign-flipping probability)**[2]**.** *Suppose that $G_t \perp\!\!\!\perp \Omega_{t-1}$ (i.e., $G_t$ and $\Omega_{t-1}$ are independent) and the distribution of $\omega_{t-1}$ is given by $P(\Omega_{t-1} = x) = \varphi_{t-1}(x)$, where $\varphi_{t-1} : \mathbb{R} \to \mathbb{R}_{\geq 0}$ is a probability density function which is symmetric w.r.t. the origin. The sign-flipping probability $P(\hat{W}_t \neq w_{t-1}|w_{t-1}, g_t)$ in the real-space update using $\eta \in [0, 1]$ is given by*

$$P(\hat{W}_t \neq w_{t-1}|w_{t-1}, g_t) = 2\int_{I_t} \varphi_{t-1}(x)dx, \quad I_t = \begin{cases} [0, \ \max(\eta g_t, 0)] & (w_{t-1} = 1) \\ [\min(\eta g_t, 0), \ 0] & (w_{t-1} = 0) \end{cases}. \quad (10)$$

Using this probability, the mask distributions satisfying the EMP/MMP are derived as follows.

**Theorem 2 (Hypermask with EMP).** *The mask distribution satisfies EMP if*

$$P(M_t = 1|w_{t-1}, g_t) = P(\hat{W}_t \neq w_{t-1}|w_{t-1}, g_t). \quad (11)$$

**Theorem 3 (Hypermask with MMP).** *The mask distribution satisfies MMP if*

$$P(M_t = 1|w_{t-1}, g_t) = \begin{cases} 1 & (P(\hat{W}_t \neq w_{t-1}|w_{t-1}, g_t) \geq \frac{1}{2}) \\ 0 & (\text{otherwise}) \end{cases}. \quad (12)$$

Theorems 2 and 3 introduce two different hypermasks. With these theorems, we have four steps to obtain a hypermask as follows: (1) Manually design a probability function $P(\omega_{t-1}) = \varphi_{t-1}(\omega_{t-1})$. (2) Compute the sign-flipping probability $P(\hat{W}_t \neq w_{t-1}|w_{t-1}, g_t)$ according to Eq. (10). (3) Compute the mask distribution according to Eq. (11) or (12). (4) Sample a hypermask $m_t$ from the distribution $P(M_t|w_{t-1}, g_t)$. The examples of modelings of $\varphi_{t-1}$ are provided and discussed in 3.3.

## 3.3 IMPLEMENTATION OF HYPERMASKS

In this subsection, we explicitly write down the probability mass functions of the hypermasks at time $t$ in the case that a manually designed distribution $P(\Omega_{t-1})$ is a Gaussian distribution $N(0, \sigma_{t-1}^2)$ where $\sigma_{t-1}$ is a standard deviation, *i.e.*, $\varphi_{t-1}(x) = \frac{1}{\sqrt{2\pi}\sigma_{t-1}}\exp(-\frac{x^2}{2\sigma_{t-1}^2})$. In the following, the previous binary weight $w_{t-1}$ and the real-valued gradient $g_t$ are supposed to be observed. First, the sign-flipping probability is computed as follows.

**Implementation 1 (Sign-flipping probability).** *Given $\omega_{t-1} \sim N(0, \sigma_{t-1}^2)$, the sign-flipping probability $P(\hat{W}_t \neq w_{t-1}|w_{t-1}, g_t)$ is calculated by Theorem 1 as*

$$P(\hat{W}_t \neq w_{t-1}|w_{t-1}, g_t) = \begin{cases} \text{erf}(\max(\tau_{t-1}g_t, 0)) & (w_{t-1} = 1) \\ -\text{erf}(\min(\tau_{t-1}g_t, 0)) & (w_{t-1} = 0) \end{cases}, \quad (13)$$

*where $\tau_{t-1} = \eta/\sqrt{2}\sigma_{t-1}$ is a temperature used in simulated annealing methods, etc.*

Using Eq. (13) and Theorem 2, the *EMP mask*, the mask satisfying the EMP, is given as follows.

---

[2]See Appendix A for the proofs.

**Implementation 2 (EMP Mask).** *Given $\omega_{t-1} \sim N(0, \sigma_{t-1}^2)$, the EMP mask is explicitly defined by*

$$P(M_t = 1 | w_{t-1}, g_t) = \begin{cases} \text{erf}(\max(\tau_{t-1} g_t, 0)) & (w_{t-1} = 1) \\ -\text{erf}(\min(\tau_{t-1} g_t, 0)) & (w_{t-1} = 0) \end{cases}. \tag{14}$$

Similarly, *MMP mask*, the mask maximizing the MP, is given by Implementation 3.

**Implementation 3 (MMP mask).** *Given $\omega_{t-1} \sim N(0, \sigma_{t-1}^2)$, the MMP mask is explicitly defined by*

$$P(M_t = 1 | w_{t-1}, g_t) = \begin{cases} [\![ g_t \geq T_{t-1}(\frac{1}{2}) ]\!] & (w_{t-1} = 1) \\ [\![ g_t \leq T_{t-1}(-\frac{1}{2}) ]\!] & (w_{t-1} = 0) \end{cases}, \quad \text{where } T_{t-1}(\cdot) = \frac{\text{erf}^{-1}(\cdot)}{\tau_{t-1}}. \tag{15}$$

The MMP mask takes 1 if the gradient going in the direction of flipping the binary weights is larger than the threshold $T_{t-1}(\pm\frac{1}{2})$.

Next, we discuss the determination of $\tau_{t-1}$ and $\sigma_{t-1}$. Assume that $P(G_t)$ is a Gaussian distribution with zero means and all elements of $\boldsymbol{\omega}_{t-1}$ and $\boldsymbol{g}_t$ are i.i.d., respectively. In the real-space update, we have $P(\Omega_t) = N(\Omega_t | 0, \sigma_{t-1}^2 + \eta^2 \hat{\sigma}_t^2)$, where $\hat{\sigma}_t^2$ is the unbiased variance of obtained $\boldsymbol{g}_t$. Under this assumption, $\sigma_t$ is updated by giving $\sigma_0$ as a hyperparameter in the following way:

$$\sigma_t = \left( \sigma_0^2 + \eta^2 \sum_{k=1}^{t} \hat{\sigma}_k^2 \right)^{\frac{1}{2}} \quad (t > 0). \tag{16}$$

With Eq. (16), the temperature is scheduled for each iteration as Implementation 4.

**Implementation 4 (Auto schedule for temperature).** *The temperature $\tau_t$ is updated by giving $\sigma_0$ and $\eta$ as the hyperparameters in the following way:*

$$\tau_t = \frac{\eta}{\sqrt{2}\sigma_t} = \frac{1}{\sqrt{2}\sqrt{(\sigma_0/\eta)^2 + \sum_{k=1}^{t} \hat{\sigma}_k^2}}. \tag{17}$$

Here, $\eta$ and $\sigma_0$ can be combined into a single hyperparameter. The temperature $\tau_t$ monotonically decreases; thus our proposed binary-space training with EMP/MMP mask converges (See Figure 5).

Finally, in addition to the above two hypermasks, we define a straightforward hypermask, the *random mask*. This hypermask uses randomly sampled values for masking.

**Implementation 5 (Random mask).** *We define the random mask by*

$$P(M_t = 1) = \delta_t, \quad \text{where } 0 \leq \delta_t \leq 1. \tag{18}$$

This definition describes that $m_t$ takes 1 with a probability $\delta_t$ regardless of whether or not the observation of $w_{t-1}, g_t$. If $\delta_t = 1$ for all iterations, the binary-space training with the random mask is equivalent to updating the binary weight $w$ with the target weight $w^*$. $\delta_t$ can be scheduled by using any learning rate scheduler such as cosine decay (Loshchilov & Hutter, 2017).

The shapes of the mask probability $P(M_t = 1 | w_{t-1}, g_t)$ for each mask are shown in Figure 3. Note that the distribution $P(\Omega_{t-1})$ can be set to other distributions. The use of a Gaussian mixture distribution as $P(\Omega_{t-1})$ could express the separation of real-valued weights observed in the late stages of BNN training in a real-space training (Rozen et al., 2022).

## 4 ANALYSIS OF MEMORY REDUCTION OF BINARY-SPACE TRAINING

In this section, we discuss memory reduction in the binary-space training of BNNs. Table 2 presents the sets of variables required simultaneously for each algorithm which result in the highest memory consumption. The table also provides the analytical values for this peak memory usage. As the table shows, our binary-space training can reduce the memory consumption relative to the number of layers $L$ by a factor of 33 compared to the real-space training, especially when $B \ll D_{hid}$.

To estimate the memory savings in a practical setting, let's consider the example of training an $L$-layered fully connected network using the MNIST. Assuming $D_{hid} = 1024$ and $B = 64$, the estimated peak memory use in binary-space training is $2.06L + 61.7$ [MiB]. In contrast, real-space training demands $34.1L - 41.4$ [MiB]. For a relatively shallow model ($L = 5$), the binary-space

Table 2: Analytical memory usage. $D_{hid}$ and $B$ are the number of hidden units and the batch size. $\Theta_{\text{train}}$ stands for a training space of weights. $O()$ is a Big O notation for space complexity w.r.t. $L$. FP was assumed to be the commonly used `float32`. The data type of $M_l$ is considered as `float32` because, when sampled, it requires a full-precision matrix to compute the mask probability.

| | $\Theta_{\text{train}}$ | Variables | | | Analytical Memory Usage [Byte] |
|---|---|---|---|---|---|
| | | `float32` | `int16` | `binary` | |
| NNs | $\mathbb{R}$ | $\{\omega_l, a_l, h_l\}_{l=1}^{L}, \nabla_{h_L}\mathcal{L}$ | - | - | $(32D_{hid} + 64B)D_{hid}L + O(1)$ |
| BNNs | $\mathbb{R}$ | $\{\omega_l\}_{l=1}^{L}, \nabla_{h_L}\mathcal{L}$ | $\{a_l\}_{l=1}^{L}$ | $\{w_l, h_l\}_{l=1}^{L}$ | $(33D_{hid} + 17B)D_{hid}L + O(1)$ |
| | $B$ | $\nabla_{W_l}\mathcal{L}, M_l$ | $\{a_l\}_{l=1}^{L}$ | $\{w_l, h_l\}_{l=1}^{L}$ | $(D_{hid} + 17B)D_{hid}L + O(1)$ |

approach consumes $\approx 55.8\%$ of what the real-space method uses. For a deeper model ($L = 50$), it uses only $\approx 9.90\%$ of the real-space's memory, resulting in $\simeq 10\text{x}$ memory saving. Note that even though the maximum memory reduction by binary space can reach up to 33x when $L \to \infty$ and $B \ll D_{hid}$, our proposed method still needs real-valued gradients, which can be the bottleneck if the number of layers is small. The additional memory savings would be achieved by quantizing the real gradients (Wang et al., 2021).

## 5 EXPERIMENTS

In this section, we show experimental results. In 5.1, we show how BNNs perform in the binary-space training compared to the real-space training. We then conduct analyses of hypermasks; specifically, we verify the validity of the EMP mask and its assumptions on a real-image dataset in 5.2.

### 5.1 EVALUATION OF PROPOSED METHODS

This experiment aims to evaluate the performance of our proposed binary-space training of BNNs relative to the real-space training of BNNs across classification tasks on three real-image datasets: MNIST (Lecun et al., 1998), CIFAR-10 (Krizhevsky, 2009), and CIFAR-100. We conducted three types of realistic evaluation: (1) Fine-tuning binary MLP header of CNN-based BNNs trained on ImageNet-1k (Deng et al., 2015). (2) Fine-tuning whole layers of CNN-based BNNs trained on ImageNet-1k. (3) Training fully connected networks from scratch. Experiments (1) and (2) are more realistic settings than (3) because training models on edge devices from scratch requires high computational cost. In (1) and (2), we used ReActNet (Liu et al., 2020) (ResNet-18 (He et al., 2016b) architecture) and AdamBNN (Liu et al., 2021a) (MobileNetV1 (Howard et al., 2017) architecture) as the backbones. In (3), we trained 4-layered binary MLP by each training method. For baseline comparisons, we utilized STE (Hubara et al., 2016), BOP (Helwegen et al., 2019) (instead of real weights, BOP keeps a real accumulator for the gradient), and ReSTE (Wu et al., 2023) for real-space training. Note that, in (1), we applied the gradient scaling of XNOR-Net (Rastegari et al., 2016) to each training method except for the naive STE. Comprehensive settings are detailed in Appendix B.

**(1) Fine-tuning Binary MLP header.** Results are summarized in Tables 3. the table shows that despite not retaining real-valued weights, our EMP Mask achieved a performance comparable to other SOTA real-space training methods for BNNs on both CIFAR-10 and larger CIFAR-100.

Table 3: Fine-tuning accuracy [%]. The backbones were trained on ImageNet-1k. In the fine-tuning, the backbones were kept fixed and 2-layered binary MLPs ($D_{hid} = 512$) were trained as headers. The best and second-best results are marked in bold with and without an underline, respectively.

| Training Method | $\Theta_{\text{train}}$ | CIFAR-10 | | CIFAR-100 | |
|---|---|---|---|---|---|
| | | ReActNet (Liu et al., 2020) | AdamBNN (Liu et al., 2021a) | ReActNet (Liu et al., 2020) | AdamBNN (Liu et al., 2021a) |
| Naive STE (Hubara et al., 2016) | | $49.09_{+0.46}$ | $84.91_{+0.35}$ | $26.19_{+0.14}$ | $61.92_{+0.24}$ |
| BOP (Helwegen et al., 2019) | $\mathbb{R}$ | $\mathbf{49.93_{+0.18}}$ | $85.16_{+0.14}$ | $\mathbf{27.01_{+0.24}}$ | $\mathbf{63.88_{+0.37}}$ |
| ReSTE (Wu et al., 2023) | | $\underline{49.47_{+0.47}}$ | $\mathbf{85.53_{+0.34}}$ | $\underline{26.31_{+0.26}}$ | $\mathbf{62.40_{+0.42}}$ |
| EMP Mask (ours) | $B$ | $\mathbf{49.73_{+0.29}}$ | $\underline{\mathbf{85.63_{+0.16}}}$ | $\mathbf{26.41_{+0.27}}$ | $62.03_{+0.39}$ |

**(2) Fine-tuning Binary CNN.** Results are summarized in Table 4. The performance difference with the real-space training was within the error range, demonstrating that our binary-space approach performs well not only in training the linear layer but also in training the convolutional layer. Additionally, the peak memory usage of our approach was only $\approx 27\%$ of that of the real-space training, making it efficient for training on edge devices.

Table 4: Fine-tuning accuracy [%]. AdamBNN was used as the backbone. Unlike Table 4, all layers, including the backbone, were trained. Memory presents the analytical memory usage described in Section 4 ($B = 128$ here).

| Method | | CIFAR-10 | CIFAR-100 |
|---|---|---|---|
| Naive STE | Accuracy [%] | $91.99_{\pm 0.35}$ | $74.87_{\pm 0.48}$ |
| | Memory [MB] | 151.1 | 151.4 |
| EMP Mask | Accuracy [%] | $91.92_{\pm 0.15}$ | $74.64_{\pm 0.23}$ |
| | Memory [MB] | 41.6 | 42.0 |

**(3) Training BNN from scratch.** Table 5 shows the results. Compared to fine-tuning, our method performs slightly worse in training a feature extractor from scratch; however as the model size expands, the performance gap narrows and the degradation of the EMP Mask was $\approx 0.4\%$ on the large MLP. Among the binary-space training methods, the EMP mask outperforms the other hypermasks across all model sizes except for the result of training the large model by the random mask using cosine decay. Interestingly, by using the cosine-decay scheduler, the random mask performed surprisingly well despite not using the gradient magnitudes; thus this result implies that our training methods could work even if the gradient is quantized to 1-bit. On the other hand, the MMP mask's performance was subpar across all models, which will be further explored in 5.2.

Table 5: Test error [%] in real- and binary-space training of 4-layered MLP from scratch on MNIST.

| | $\Theta_{\text{train}}$ | Training Method | Number of hidden units $D_{hid}$ | | |
|---|---|---|---|---|---|
| | | | 128 (*Small*) | 2048 (*Medium*) | 8192 (*Large*) |
| NN | $\mathbb{R}$ | Backpropagation (SGD) | $2.18_{\pm 0.17}$ | $1.37_{\pm 0.05}$ | $1.46_{\pm 0.03}$ |
| | $\mathbb{R}$ | Naive STE | $5.81_{\pm 0.23}$ | $2.10_{\pm 0.05}$ | $2.04_{\pm 0.07}$ |
| BNN | $B$ | Random mask, $\delta_t = 1$ | $93.02_{\pm 1.91}$ | $89.46_{\pm 4.29}$ | $86.70_{\pm 5.40}$ |
| | | Random mask, $\delta_t = 10^{-3}$ | $9.37_{\pm 0.12}$ | $3.55_{\pm 0.09}$ | $4.02_{\pm 0.06}$ |
| | | Random mask, cosine decay | $\mathbf{9.20_{\pm 0.24}}$ | $\mathbf{3.04_{\pm 0.03}}$ | $\mathbf{2.44_{\pm 0.04}}$ |
| | | MMP mask | $51.32_{\pm 2.43}$ | $22.34_{\pm 0.78}$ | $18.02_{\pm 0.31}$ |
| | | EMP mask | $\mathbf{6.82_{\pm 0.09}}$ | $\mathbf{2.95_{\pm 0.05}}$ | $\mathbf{2.50_{\pm 0.04}}$ |

## 5.2 VERIFICATION OF EMP MASK AND ITS ASSUMPTIONS

**Verification of EMP mask.** We empirically demonstrate that the EMP mask achieves the EMP. To substantiate this, we compare the mask probability $P(M_t = 1 | w_{t-1}, g_t)$ and the sign-flipping probability $P(\hat{W}_t \neq w_{t-1} | w_{t-1}, g_t)$ in *real-space* training. If these probabilities align for all $w_{t-1}$ and $g_t$, the EMP is satisfied as per Theorem 2. Figure 3 displays the results, juxtaposing the mask probabilities of hypermasks with the sign-flipping probabilities of the input layer in a small FCN trained on MNIST. Note that we used the actual standard deviations of the real weights as $\sigma_{t-1}$ and $w_{t-1} = 1$ for the mask probabilities. As illustrated on the left, there is a notable overlap between the EMP mask's probability and the sign-flipping probability, *i.e.*, the EMP can be satisfied in binary-

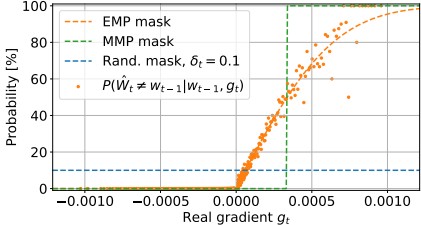 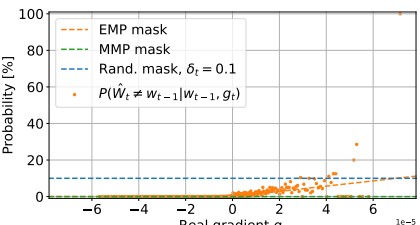

Figure 3: Mask probability $P(M_t = 1 | w_{t-1} = 1, g_t)$ and sign-flipping probability at the first (left) and second (right) epochs in the real-space training. The sign-flipping probability was calculated by dividing the gradient into several intervals and tabulating the sign-flipping rates of the real weights.

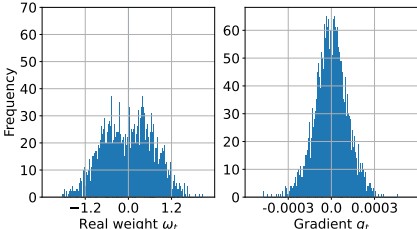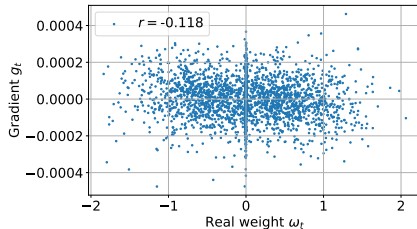

Figure 4: (Left) Real weight and gradient distributions of the output layer at the 10th epoch of the real-space training. (Right) Joint distribution of these variables ($r$ is the correlation coefficient).

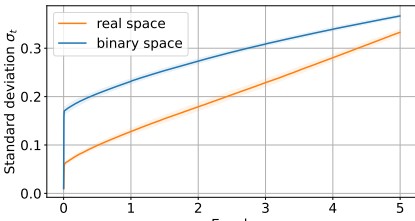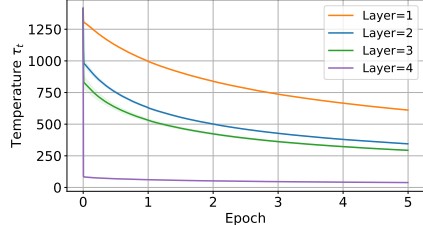

Figure 5: (Left) Standard deviations of the real weight of the output layer in the real- and binary-space training. Both used the same learning rate. In the binary-space training, the standard deviation is updated by Eq. (16). (Right) Auto temperature schedule of all layers in the binary space training.

space training. As training advances, the peak value of the sign-flipping probability diminishes. By the second epoch, no probability exceeds 50%. Consequently, the MMP mask consistently takes 0, leading to underfitting in the binary-space training of BNNs, as evident in Table 5.

**Verification of assumptions.** During the binary-space training, certain assumptions were made to calculate the sign-flipping probability of the binary weights, *e.g.*, $\Omega_{t-1} \perp\!\!\!\perp G_t$, $G_t \sim N(0, \hat{g}_t^2)$ and $\Omega_{t-1} \sim N(0, \sigma_{t-1}^2)$. The left side of Figure 4 displays the distributions of the real-valued weights and gradient at the 10-th epoch during the real-space training of a small FCN on MNIST. As depicted, the real-valued weight distribution initialized by the Gaussian distribution basically preserves its form, and the gradient also obeys a zero-mean Gaussian distribution. The joint distribution is shown in the right of Figure 4, and the correlation coefficient of $r = -0.118$ suggests near-independence between the distributions. However, in BNNs, these distributions might deviate from the Gaussian form as the training progresses (See Appendix C). Moreover, the correlation coefficient is small but not negligible. These facts could prevent the binary-space training from accurately emulating the real-space training. Figure 5 shows the standard deviation of the hypothetical distribution of the real-valued weight in the binary-space training of this setting, and it is slightly different from that of the actual distribution in the real-space training. A more precise assumption regarding distribution, coupled with a deeper exploration of the dependence between real-valued weights and gradients, could enhance the fidelity of binary-space training emulation and further refine our method.

## 6 DISCUSSION

Prior work largely trained low-precision weights through gradient quantization or model-based discrete optimization like MIP. This study's major contribution is the introduction of a novel approach for directly training ultra-low precision 1-bit weights. This method yielded performance comparable to BNNs trained with real-valued weights using traditional real-space SGD. Also, our approach has a computational complexity nearly identical to SGD, alleviating the challenges of MIP-based methods on large datasets.

Although our method successfully achieved the comparable performance to the previous method while eliminating real weights, our binary-space approach still uses real gradients to estimate the probability of binary weight sign-flipping, representing the memory consumption bottleneck during training. Quantizing these real-valued gradients has been recognized as a computationally effective strategy in BNNs and there are some previous works (Zhou et al., 2016; Wang et al., 2021). Incorporating these quantization methods into our binary-space training would alleviate this bottleneck.

**Ethics statement.** Our proposed binary-space training for BNNs is anticipated to facilitate the localization and personalization of AI systems on low-end edge devices. Nevertheless, excessive bias in the data employed for local training may yield unforeseen device behavior, and to circumvent this, it may be imperative to devise a method to assess the validity of local learning.

**Reproducibility statement.** To reproduce the results of the experiments, the source code and hyperparameters used in the experiments are included in the supplementary material. See Appendix B and the attached code.

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

## A    PROOF OF THEOREMS

**Proof of Theorem 1.** Assume that $\Omega_{t-1} \perp\!\!\!\perp G_t$, then we have

$$
\begin{aligned}
p(\hat{w}_t|w_{t-1}, g_t) &= \frac{p(\hat{w}_t, w_{t-1}, g_t)}{p(w_{t-1}, g_t)} \\
&= \int_{-\infty}^{\infty}\int_{-\infty}^{\infty} \frac{p(\hat{w}_t, w_{t-1}, g_t, \omega_t, \omega_{t-1})}{p(w_{t-1}, g_t)} d\omega_t d\omega_{t-1} \\
&= \int_{-\infty}^{\infty}\int_{-\infty}^{\infty} \frac{p(\omega_t|\omega_{t-1}, g_t)p(\hat{w}_t|\omega_t)p(w_{t-1}|\omega_{t-1})p(\omega_{t-1})p(g_t)}{p(w_{t-1})p(g_t)} d\omega_t d\omega_{t-1} \\
&= \int_{-\infty}^{\infty}\int_{-\infty}^{\infty} p(\omega_t|\omega_{t-1}, g_t)p(\hat{w}_t|\omega_t)p(\omega_{t-1}|w_{t-1}) d\omega_t d\omega_{t-1}.
\end{aligned}
\tag{19}
$$

For simplicity, we consider the case where $w_{t-1} = 1$ and $\hat{w}_t = 0$ in the following. In this case, the following two equations hold:

$$
p(\hat{w}_t = 0|\omega_t) = \begin{cases} 1 & (\omega_t < 0) \\ 0 & (\omega_t \geq 0) \end{cases},
\tag{20}
$$

$$
p(\omega_{t-1}|w_{t-1} = 1) = \begin{cases} 2\varphi(\omega_{t-1}) & (\omega_{t-1} \geq 0) \\ 0 & (\omega_{t-1} < 0) \end{cases}.
\tag{21}
$$

By substituting Eqs. (20-21) into Eq. (19), we have the following expression:

$$
p(\hat{w}_t = 0|w_{t-1} = 1, g_t) = 2\int_0^{\infty} \varphi(\omega_{t-1}) \left( \int_{-\infty}^0 p(\omega_t|\omega_{t-1}, g_t)d\omega_t \right) d\omega_{t-1}.
\tag{22}
$$

Considering the update rule of $\omega_t$ (Eq. (4)), we have

$$
p(\omega_t|\omega_{t-1}, g_t) = \delta(\omega_{t-1} - \eta g_t - \omega_t),
\tag{23}
$$

where $\delta$ is a Dirac's delta. It's easy to check that

$$
\int_{-\infty}^0 p(\omega_t|\omega_{t-1}, g_t)d\omega_t = \begin{cases} 1 & (\omega_{t-1} < \eta g_t) \\ 0 & (\text{otherwise}) \end{cases},
\tag{24}
$$

and by substituting Eq. (24) into Eq. (22), we finally obtain the following equality:

$$
p(\hat{w}_t \neq w_{t-1}|w_{t-1} = 1, g_t) = 2\int_0^{\max(\eta g_t, 0)} \varphi(\omega_{t-1})d\omega_{t-1}.
\tag{25}
$$

The same derivation can also be done for the another case $w_{t-1} = 0$. From the above, Eq. (13) has been proved.

**Proof of Theorem 2.** The EMP is equivalent to

$$
\forall w_{t-1}, g_t \ \ P(W_t = 1|w_{t-1}, g_t) = P(\hat{W}_t = 1|w_{t-1}, g_t).
\tag{26}
$$

Also, Table 6 shows the four patterns of the combination of $w_{t-1}$ and $g_t$. From the table, Eq. (26) is equivalent to

$$
\begin{aligned}
P(\overline{M}_t = 1|w_{t-1} = 1, g_t \geq 0) &= P(\hat{W}_t = 1|w_{t-1} = 1, g_t \geq 0) \\
\wedge P(M_t = 1|w_{t-1} = 0, g_t < 0) &= P(\hat{W}_t = 1|w_{t-1} = 0, g_t < 0)
\end{aligned}
\tag{27}
$$

Thus, the EMP is satisfied if and only if the mask probability satisfies

$$
\begin{aligned}
P(M_t = 1|w_{t-1}, g_t) &= P(\hat{W}_t \neq w_{t-1}|w_{t-1}, g_t) \\
&((w_{t-1} = 1 \wedge g_t \geq 0) \vee (w_{t-1} = 0 \wedge g_t < 0))
\end{aligned}
\tag{28}
$$

The EMP mask obviously satisfies Eq. (28) and then the theorem is proved.

**Proof of Theorem 3.** The MP is equivalent to

$$\forall w_{t-1}, g_t \; MP$$
$$= P(W_t = 1|w_{t-1}, g_t)P(\hat{W}_t = 1|w_{t-1}, g_t)$$
$$+ (1 - P(W_t = 1|w_{t-1}, g_t))(1 - P(\hat{W}_t = 1|w_{t-1}, g_t))$$
$$= (2P(\hat{W}_t = 1|w_{t-1}, g_t) - 1)P(W_t = 1|w_{t-1}, g_t)$$
$$+ 1 - P(\hat{W}_t = 1|w_{t-1}, g_t). \tag{29}$$

Table 6: Values of $W_t$ in each $w_{t-1}$ and $g_t$ combination.

| $w_{t-1}$ | 1 | 1 | 0 | 0 |
|---|---|---|---|---|
| $g_t$ | + | − | + | − |
| $W_t$ | $\overline{M}_t$ | 1 | 0 | $M_t$ |

The probability $P(W_t = 1|w_{t-1}, g_t)$ maximizes MP if and only if

$$\left( P(\hat{W}_t = 1|w_{t-1}, g_t) \geq \frac{1}{2} \Rightarrow P(W_t = 1|w_{t-1}, g_t) = 1 \right) \tag{30}$$
$$\wedge \left( P(\hat{W}_t = 1|w_{t-1}, g_t) < \frac{1}{2} \Rightarrow P(W_t = 1|w_{t-1}, g_t) = 0 \right).$$

Using Table 6, it is easy to show that Theorem 3 holds as well as Theorem 2.

## B    DETAILED SETTINGS OF EXPERIMENTS

This section reports the detailed settings of the experiments in Section 5. For the experiments, 4 GPUs (Tesla P100-SXM2-16GB) with 56 CPU cores were used.

### B.1    HYPERPARAMETERS

**(1) Fine-tuning Binary MLP header.** We used ReActNet (Liu et al., 2020) (ResNet-18 (He et al., 2016b) architecture) and AdamBNN (Liu et al., 2021a) (MobileNetV1 (Howard et al., 2017) architecture) as the backbones. The models were trained on ImageNet-1k (Deng et al., 2015) and we used the pre-trained weights available on GitHub. Only in this experiment, we applied the gradient scaling of XNOR-Net (Rastegari et al., 2016) to each training method except for the naive STE. The binary header is 2-layer MLP and the number of the hidden units were set to 512. The numbers of the training epochs were 500 on CIFAR-10 and 250 on CIFAR-100. The batch size was 32,768.

**(2) Fine-tuning Binary CNN.** We used ReActNet (Liu et al., 2020) (ResNet-18 (He et al., 2016b) architecture) and AdamBNN (Liu et al., 2021a) (MobileNetV1 (Howard et al., 2017) architecture) as the pre-trained model. The models were trained on ImageNet-1k (Deng et al., 2015) and we used the pre-trained weights available on GitHub. These pre-trained models were trained in the real- and binary-space. The numbers of the training epochs were 100 on CIFAR-10 and CIFAR-100. The batch size was 128.

**(3) Training BNN from scratch.** In this experiment, for all training algorithms, we initialized the real-valued weights by random values sampled from Gaussian distribution $N(0, 0.01^2)$ and the binary-valued weights by random values sampled from Bernoulli distribution $B(p = 0.5)$. The standard deviation $\sigma_0$ o the distribution of the real-valued weight in the binary-space training was assumed to be $0.01/\eta$. The batch size was 16,384. The numbers of the training epochs were 1,000 on Digit Dataset, 2,000 on MNIST, 500 on CIFAR-10, and 100 on Tiny ImageNet.

Table 7 shows the search sets of the learning rate $\eta$ for the real- and binary-space training and a probability $\delta$ for the random mask. These search sets were determined by a rough tuning of 10-fold intervals.

Table 7: Search sets of hyperparameters used in 5.1 for all dataset. The search set of learning rate is defined as $\{\text{round}(10^{x/10}) : x = 0, ..., 10\}$.

| Hyperparameter | Search Set |
|---|---|
| Learning rate $\eta$ | $\{1.0, 1.3, 1.6, 2.0, 2.5, 3.2, 4.0, 5.0, 6.3, 7.9, 10.0\}$ |
| Delta $\delta$ | $\{10^{-1}, 10^{-2}, 10^{-3}, 10^{-4}\}$ |

The best hyperparameters used in Table 5 are shown in Table 8, respectively. To find these best hyperparameters, a grid search was used; 10% of the training data were used as a validation set, and the hyperparameter which obtained the best validation performance at the 100th epoch was chosen as the best hyperparameter. The values of hyperparameters for the real-space training were set equal to those for the binary-space training. This is because learning rate works the same in real- and binary-space training.

Table 8: Best hyperparameters used in Table 5. We used a different search set for MMP mask.

| $D_{hid}$ | Hyperparameter | STE | EMP mask | MMP Mask | Random mask (cosine decay) |
|---|---|---|---|---|---|
| 128 | Learning Rate $\eta$ | 10.0 | 10.0 | 63 | - |
| | Delta $\delta$ | - | - | - | $10^{-3}$ |
| 2048 | Learning Rate $\eta$ | 7.9 | 7.9 | 63 | - |
| | Delta $\delta$ | - | - | - | $10^{-3}$ |
| 8192 | Learning Rate $\eta$ | 10.0 | 10.0 | 63 | - |
| | Delta $\delta$ | - | - | - | $10^{-3}$ |

**Experiments 5.2** In these experiments, we used the small FCN ($D_{hid} = 256$) as the training model and the MNIST as the dataset.

Figure 3 shows the mask and sign-flipping distributions of the input layer in the real-space training. We used $\eta = 20$ as the learning rate and $\sigma_0 = 0.01$ as the initial value of the standard deviation of the real-valued weight.

Figure 4 shows the distributions of the real-valued weight and gradient (left) and the joint distribution of them (right) of the output layer in the real-space training. We used $\eta = 20$ as the learning rate and $\sigma_0 = 0.01$ as the initial value of the standard deviation of the real-valued weight.

Figure 5 shows the standard deviations of the real-valued weight of the output layer in the real- and binary-space training (left) and the temperature in the binary-space training (right). In the left figure, the standard deviation in the binary-space training is that of the hypothetical distribution $P(\Omega_t)$ which is updated by Eq. (16). The learning rates of the real- and binary-space training were set to the same value $\eta = 20$. In the right figure, the same learning rate $\eta = 20$ was used.

## C   DISTRIBUTIONS OF THE REAL-VALUED WEIGHT AND GRADIENT

In this section, we provide the additional analysis of the distributions of the real-valued weight and gradient in addition to 5.2. Here, we used the small ($D_{hid} = 256$) and large ($D_{hid} = 8192$) FCNs as the training models and the MNIST as the dataset.

### C.1   WEIGHT DISTRIBUTION

In the construction of the hypermasks, we assumed a Gaussian distribution as the prior distribution of real-valued weights. Here, we validate the validity of this assumption in addition to 5.2. Figures 8 and 9 show the real-valued weight distribution of BNNs trained in the real space. As the figures show, in the small BNN, it is observed that the real-valued weight distribution, initialized with a Gaussian distribution, diverges from the Gaussian distribution over time as training progresses, with the number of elements near zero increasing. This phenomenon is particularly pronounced in the output layer. On the other hand, in the large BNN, deviations from the Gaussian distribution occur almost exclusively in the output layer, and even there, the deviation is minimal compared to that in the small model. The size-dependent performance improvement observed in the binary-space training could potentially be attributed to the extent of this distribution collapse.

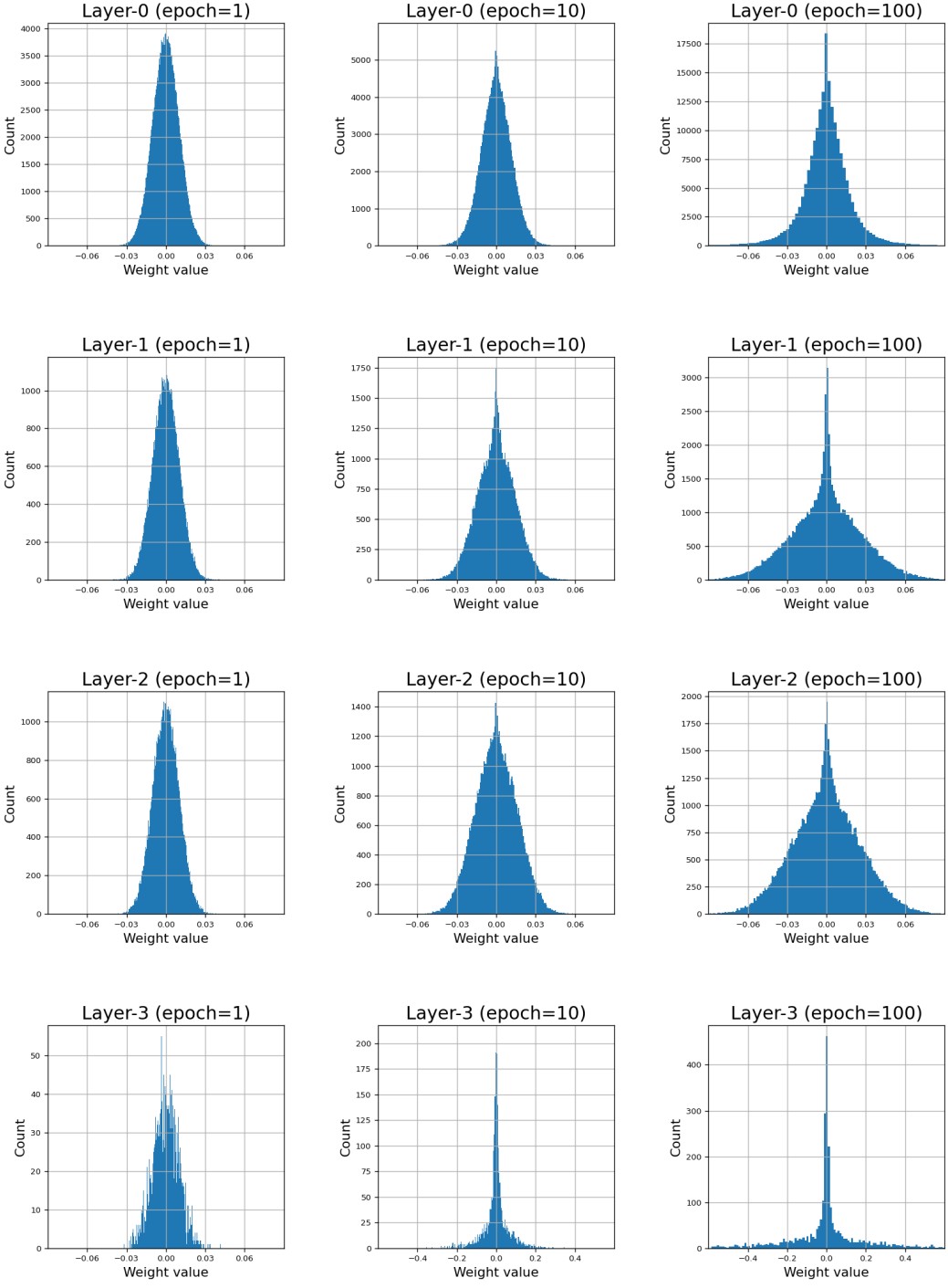

Figure 6: Weight distributions of the smaller BNN on MNIST.

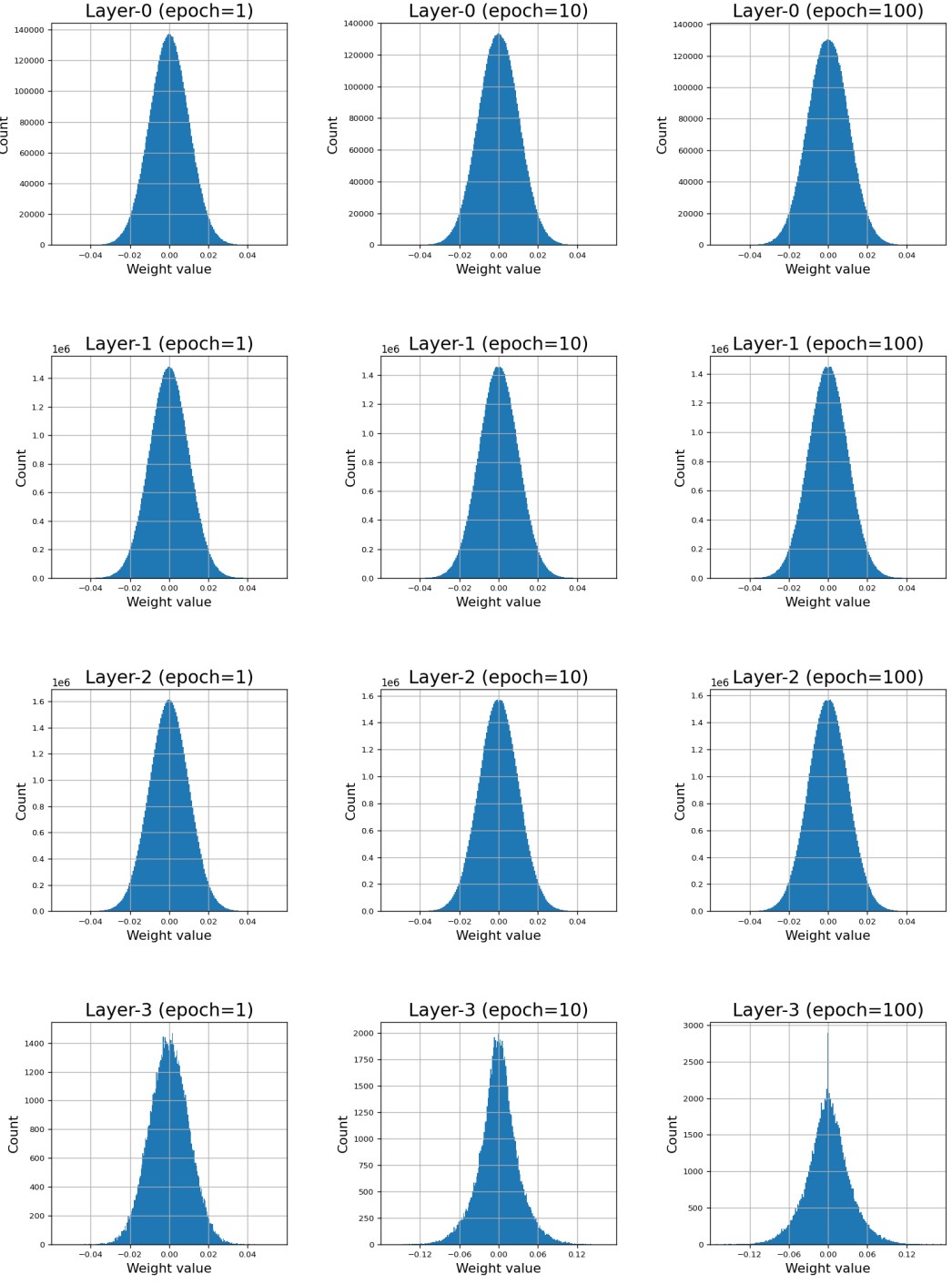

Figure 7: Weight distributions of the larger BNN on MNIST.

### C.2 GRADIENT DISTRIBUTIONS

Next, we confirm the validity of the approximation of the real-valued gradient by a Gaussian distribution during the update of $\tau_t$. Figures 8 and 9 show the distribution of real-valued gradients. As can be seen from the figures, from the second layer onwards, the distribution takes a shape similar to a Gaussian distribution at the initial stage of learning, but as learning progresses, the distribution in the upper layers becomes multi-modal, diverging from the Gaussian distribution. This is thought to be due to the fact that as the realizations of the binary weights become limited in the upper layers as the training progresses, the realizations of the gradients are also limited. In the first layer, there are significantly many values near zero, deviating from the shape of the Gaussian distribution from the initial stage of learning. Reflecting these trends to the approximation of the real-valued gradient could potentially improve the binary-space training of BNNs.

## D  PERFORMANCE UNDER VARIOUS SETTINGS

In this section, we evaluate the proposed hypermasks under settings different from those in the main paper. Specifically, we experimentally investigate the changes in performance when the dimension of the hidden layers and batch size. We used the MNIST as the dataset.

**Number of hidden units** We investigate the performance improvement of the proposed method when the number of hidden units is increased while keeping the batch size constant ($B = 16, 384$). The number of epochs was set to 2,000. Table 9 and Figure 10 show the results. As these results show, in the proposed methods, the degradation form the real-space training to the binary-space training decreases as the number of the hidden units increases, and further performance improvement can be expected by increasing the dimension from 8192.

**Batch size** Next, we investigate the performance of the proposed method when the number of batch size is increased while keeping the number of the hidden units constant ($D_{hid} = 1024$). The number of epochs was set to 200. The results are in Table 10 and Figure 11. As these results demonstrate, the performance of the proposed methods become better as the batch size increases. One possible reason for this is that, in training in a real space, real-valued weights can memorize local optima for multiple batches, whereas in a binary space, memorization is difficult since the 1-bit weight can retain less information.

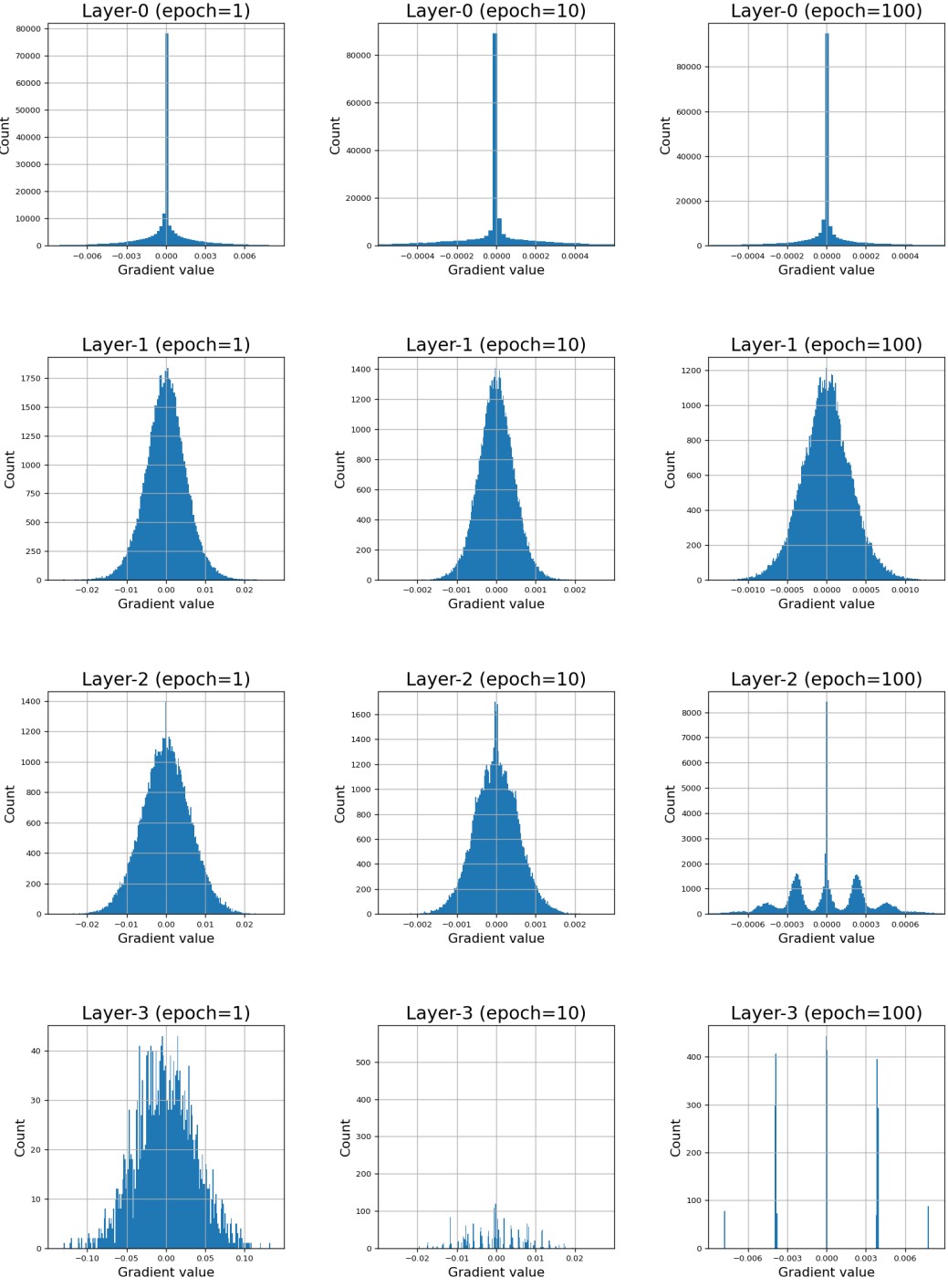

Figure 8: Gradient distributions of the smaller BNN on MNIST.

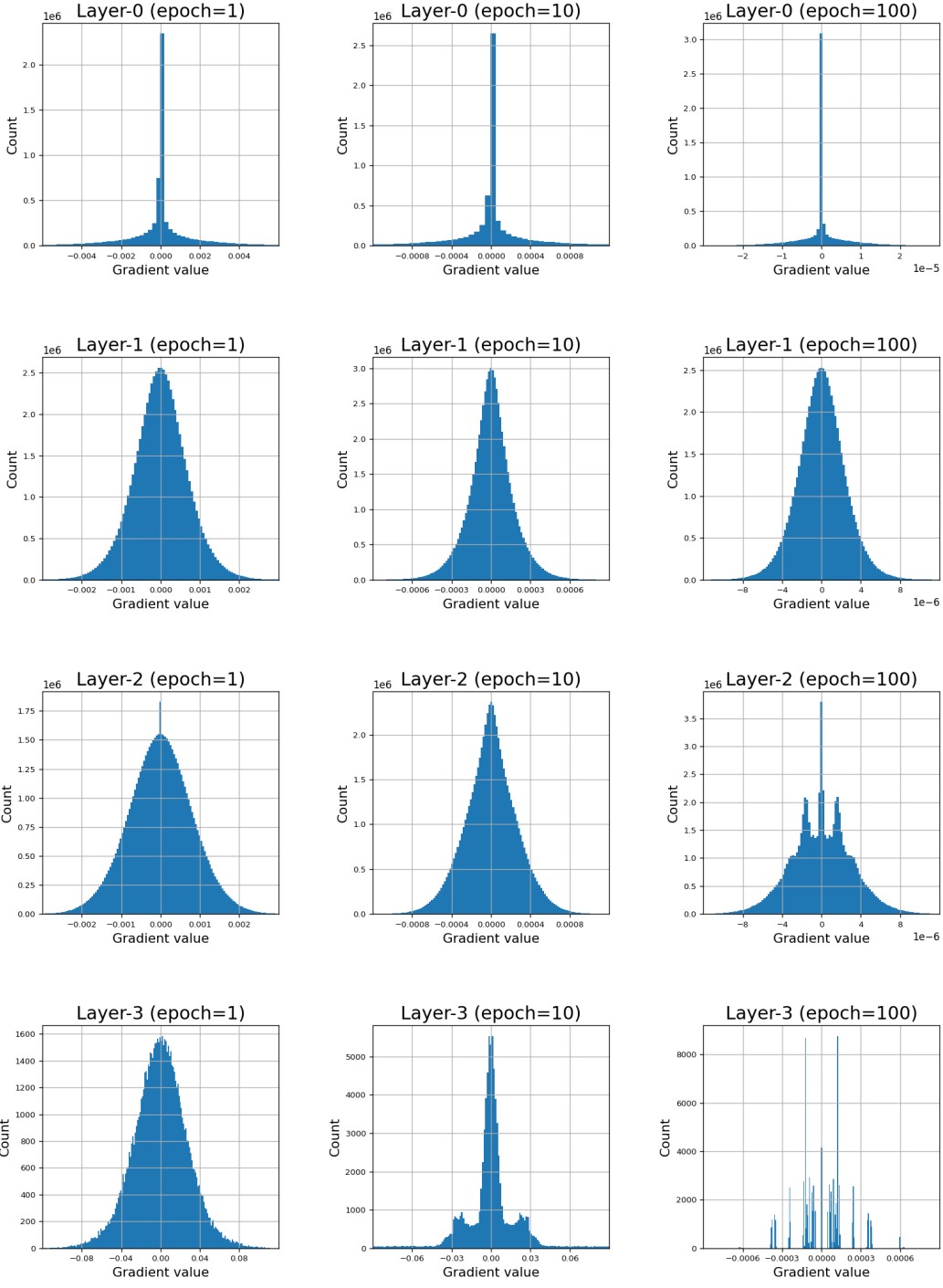

Figure 9: Gradient distributions of the larger BNN on MNIST.

Table 9: Test errors (top-1) [%] on MNIST with the mean and standard deviation over five different seeds. The batch size and the number of epochs were set to 16,384 and 2,000, respectively.

| $D_{hid}$ | Training Algorithm | | |
| --- | --- | --- | --- |
| | SGD + BP | SGD + STE | EMP mask |
| 128 | $2.18_{\pm 0.17}$ | $5.81_{\pm 0.23}$ | $6.82_{\pm 0.09}$ |
| 256 | $1.83_{\pm 0.08}$ | $3.81_{\pm 0.12}$ | $5.05_{\pm 0.09}$ |
| 512 | $1.64_{\pm 0.05}$ | $2.61_{\pm 0.07}$ | $4.04_{\pm 0.11}$ |
| 1024 | $1.46_{\pm 0.08}$ | $2.31_{\pm 0.08}$ | $3.33_{\pm 0.05}$ |
| 2048 | $1.37_{\pm 0.05}$ | $2.10_{\pm 0.05}$ | $2.95_{\pm 0.05}$ |
| 4096 | $1.37_{\pm 0.07}$ | $2.10_{\pm 0.07}$ | $2.65_{\pm 0.06}$ |
| 8192 | $1.46_{\pm 0.03}$ | $2.04_{\pm 0.07}$ | $2.50_{\pm 0.04}$ |

Table 10: Test errors (top-1) [%] on MNIST with the mean and standard deviation over five different seeds. The number of the hidden units and the number of epochs were set to 2,000 and 200, respectively.

| $B$ | Training Algorithm | |
| --- | --- | --- |
| | SGD + STE | EMP mask |
| 128 | $2.06_{\pm 0.10}$ | $9.42_{\pm 0.17}$ |
| 256 | $2.21_{\pm 0.33}$ | $8.04_{\pm 0.09}$ |
| 512 | $2.18_{\pm 0.18}$ | $6.94_{\pm 0.07}$ |
| 1024 | $2.00_{\pm 0.14}$ | $5.66_{\pm 0.08}$ |
| 2048 | $2.68_{\pm 0.09}$ | $4.82_{\pm 0.08}$ |
| 4096 | $2.66_{\pm 0.08}$ | $4.22_{\pm 0.10}$ |
| 8192 | $3.09_{\pm 0.08}$ | $3.90_{\pm 0.12}$ |

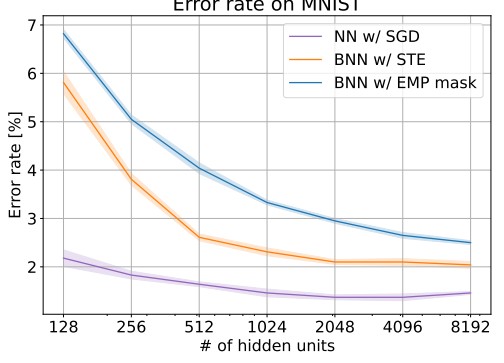

Figure 10: Change in test error [%] as number of hidden units increases.

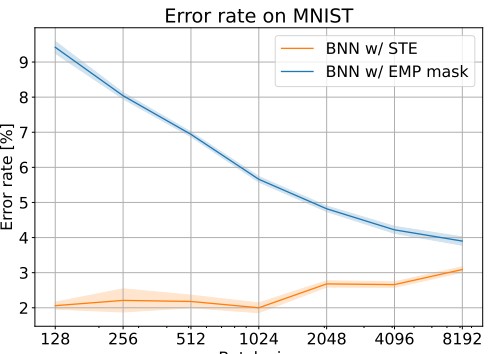

Figure 11: Change in test error [%] as batch size increases.

## E  HYPERMASK TO MINIMIZE WEIGHT UPDATE ANGLE

Here, we investigate a hypermask that explicitly minimizes the angle between the real-valued gradient $\boldsymbol{g}_t$ and the update direction of binary weights $\boldsymbol{w}_t - \boldsymbol{w}_{t-1}$, where $\boldsymbol{g}_t \in \mathbb{R}^N$ and $\boldsymbol{w}_t, \boldsymbol{w}_{t-1} \in B^N = \{0, 1\}^N$. This hypermask is formulated as

$$\boldsymbol{m}_t^* = \operatorname{argmin}_{\boldsymbol{m}_t} \angle \left(-\boldsymbol{g}_t, \boldsymbol{w}_t - \boldsymbol{w}_{t-1}\right). \tag{31}$$

**Proposition.** *Define* $I_1 = \{i \mid w_{t-1,i} = 1, [\![-g_{t,i} > 0]\!] = 0\}, I_2 = \{i \mid w_{t-1,i} = 0, [\![-g_{t,i} > 0]\!] = 1\}, I = I_1 \cup I_2$ *and* $\tilde{g}_t$ *by*

$$\tilde{g}_{t,i} = \begin{cases} g_{t,i} & (i \in I) \\ 0 & (i \notin I) \end{cases}, \tag{32}$$

*then, the optimal hypermask* $\boldsymbol{m}_t^*$ *is given by*

$$\boldsymbol{m}_t^* = \operatorname*{argmin}_{\boldsymbol{m}_t} \angle (\operatorname{abs}(\tilde{\boldsymbol{g}}_t), \boldsymbol{m}_t) \quad s.t. \quad m_{t,i} = 0 \ (i \notin I) \tag{33}$$

*Proof.* For the update direction of binary weights, the following equality holds:

$$\begin{aligned} \boldsymbol{w}_t - \boldsymbol{w}_{t-1} &= (\overline{\boldsymbol{m}}_t \boldsymbol{w}_{t-1} \vee \boldsymbol{m}_t [\![-\boldsymbol{g}_t \geq 0]\!]) - \boldsymbol{w}_{t-1} \\ &= -\boldsymbol{m}_t \odot (\boldsymbol{w}_{t-1} - [\![-\boldsymbol{g}_t > 0]\!]), \end{aligned} \tag{34}$$

where $\odot$ is an element-wise multiplication (Hadamard product). From the monotonicity of $\cos^{-1}(\cdot)$, we have

$$\begin{aligned} &\operatorname*{argmin}_{\boldsymbol{m}_t} \angle (\boldsymbol{g}_t, \boldsymbol{m}_t \odot (\boldsymbol{w}_{t-1} - [\![-\boldsymbol{g}_t > 0]\!]))) \\ &= \operatorname*{argmin}_{\boldsymbol{m}_t} \cos^{-1} \left( \frac{\sum_{i \in I_1} g_{t,i} m_{t,i} + \sum_{i \in I_2} g_{t,i}(-m_{t,i})}{\sqrt{\sum_{i=1}^N g_{t,i}^2} \sqrt{\sum_{i \in I_1} m_{t,i}^2 + \sum_{i \in I_2} (-m_{t,i})^2}} \right) \\ &= \operatorname*{argmin}_{\boldsymbol{m}_t} \cos^{-1} \left( \sqrt{\frac{\sum_{i \in I} |g_{t,i}|^2}{\sum_{i=1}^N g_{t,i}^2}} \times \frac{\sum_{i \in I} |g_{t,i}| m_{t,i}}{\sqrt{\left(\sum_{i \in I} |g_{t,i}|^2\right) \left(\sum_{i \in I} m_{t,i}^2\right)}} \right) \\ &= \operatorname*{argmin}_{\boldsymbol{m}_t} \cos^{-1} \left( \frac{\sum_{i \in I} |g_{t,i}| m_{t,i}}{\sqrt{\left(\sum_{i \in I} |g_{t,i}|^2\right) \left(\sum_{i \in I} m_{t,i}^2\right)}} \right). \end{aligned} \tag{35}$$

Here, let $m_{t,i}$ be fixed with 0 if $i \notin I$, then we obtain the following equation:

$$\begin{aligned} \operatorname*{argmin}_{\boldsymbol{m}_t} \angle (-\boldsymbol{g}_t, \boldsymbol{w}_t - \boldsymbol{w}_{t-1}) &= \operatorname*{argmin}_{\boldsymbol{m}_t} \cos^{-1} \left( \frac{\operatorname{abs}(\tilde{\boldsymbol{g}}_t) \cdot \boldsymbol{m}_t}{\|\operatorname{abs}(\tilde{\boldsymbol{g}}_t)\|_2 \|\boldsymbol{m}_t\|_2} \right) \\ &= \operatorname*{argmin}_{\boldsymbol{m}_t} \angle (\operatorname{abs}(\tilde{\boldsymbol{g}}_t), \boldsymbol{m}_t) \end{aligned} \tag{36}$$

The proposition is thus proved. $\qquad\square$

To obtain approximate solutions of Eq. (33), we give the following heuristic mask distribution

$$P(m_{t,i} = 1 | \boldsymbol{w}_{t-1}, \boldsymbol{g}_t) = \begin{cases} \beta_t & (\operatorname{abs}(\tilde{g}_{t,i}) > \gamma_t \sigma_{g_t}) \\ \alpha_t & (0 < \operatorname{abs}(\tilde{g}_{t,i}) \leq \gamma_t \sigma_{g_t}) \\ 0 & (\operatorname{abs}(\tilde{g}_{t,i}) = 0) \end{cases}. \tag{37}$$

If $\beta_t = \delta_t, \alpha_t = 0, \gamma_t = 0$, the definition in Eq. (37) is equivalent to that of the random hypermask (Eq. (18)).

