# OpenReview forum: "Training Binary Neural Networks in a Binary Weight Space"
_ICLR.cc/2024/Conference — Submitted to ICLR 2024_

### Official Review · Reviewer_2PTe · 2023-10-19

**Soundness:** 3 good
**Presentation:** 2 fair
**Contribution:** 3 good
**Rating:** 3
**Confidence:** 5

**Summary:**

The paper trains BNNs witthout real-valued weights to save memory. They define an update probability for binary weights, determined by the current binary weights and real-valued gradients. The binary weights generated by the method match those obtained by SGD in the real-space training of BNNs in the expectation.

**Strengths:**

The paper proposes a new method to train BNNs without using real-valued weights, which is novel.

The theoretical analysis of the paper is sufficient.

**Weaknesses:**

The experimental results in Tab.3 seem extremely bad on medium-sized datasets such as CIFAR-10 and Tiny-ImageNet, for both SGD+STE method and the proposed method. This raises the question of whether it is reasonable to abandon the real-valued weights when training BNNs. All the previous and proposed methods sacrifice too much classification accuracy for training efficiency.

The author claims that this training strategy is useful for edge devices. However, people seldom train models on edge devices from scratch. Thus, it is more reasonable for the author to conduct experiments on finetuning a pre-trained BNN model with binary weights only.

Since the results given in the paper are far from satisfactory, it is hard to judge the effectiveness of the proposed method.

You still have real-valued gradients during training, how could you reduce memory usage by 33? Besides, memory consumption should be listed in the table.

**Questions:**

See weaknesses above.

---

> ### Author Response · Authors · 2023-11-20
> **Response to the reviewer 2PTe**
>
> We thank the reviewer for suggesting an experimental design to better evaluate our method. We have answered each of the reviewer's questions below.
>
> > The experimental results in Tab.3 seem extremely bad on medium-sized datasets such as CIFAR-10 and Tiny-ImageNet, for both SGD+STE method and the proposed method. This raises the question of whether it is reasonable to abandon the real-valued weights when training BNNs.
>
> The underperformance of BNNs trained on CIFAR-10 and Tiny-ImageNet (shown in Table 3 of the submitted version) is mainly due to the limited generalization ability of the 4-layer MLP, not the proposed training algorithm. To show that our method (or STE) has a capability to achieve satisfactory performance on CIFAR-10 and the larger CIFAR-100, we conducted additional experiments using AdamBNN (MobileNetV1 architecture) and ReActNet (ResNet-18 architecture), which are detailed in the global response. In summary of the global response, our method achieved a top-1 accuracy of 91.94% on CIFAR-10 and 74.64% on CIFAR-100, demonstrating that our binary-space approach performs sufficiently on these medium-sized datasets.
>
> > The author claims that this training strategy is useful for edge devices. However, people seldom train models on edge devices from scratch. Thus, it is more reasonable for the author to conduct experiments on fine-tuning a pre-trained BNN model with binary weights only.
>
> We thank the reviewer for suggesting a more practical experimental setting. We conducted experiments where binary models are pre-trained with ImageNet-1k on a central server and are fine-tuned on a medium-sized dataset obtained on an edge device. The detailed settings and results have been described in the global response.
>
> > Since the results given in the paper are far from satisfactory, it is hard to judge the effectiveness of the proposed method.
>
> The underfittings were mainly due to the limited generalization performance of the 4-layer MLP, so we experimented with CNN-based BNNs with more expressive power, and have reported the results in the global response. In the experiments, our method achieved a top-1 accuracy of 91.94% on CIFAR-10 and this accuracy is close to the state-of-the-art in BNN training algorithms.
>
> > You still have real-valued gradients during training, how could you reduce memory usage by 33x Besides, memory consumption should be listed in the table.
>
> The fact that our method can achieve a memory reduction of 33x has been described in detail in Section 4. Simply stated, the proportionality coefficients of the maximum memory usage for binary- and real-space training w.r.t. the number of layers L are (D_{hid} + 17B)D_{hid} and (33D_{hid} + 17B)D_{hid}, and the memory usage that is not proportional to L can be ignored when L is sufficiently large. Thus, if L is large and D_{hid}>>B, our binary-space training can achieve 33x memory reduction. However, as already mentioned in Section 4, this is not a realistic setting, and in training 4-layer MLPs on MNIST, the memory reduction is 2x due to the real-valued gradient bottleneck. In order not to confuse the reader, we have revised the introduction and Section 4 to clarify the 33x memory reduction is asymptotic analysis. Additionally, in response to the points raised, we have reported the maximum memory consumption in the additional experiments in the global response. In the training of AdamBNN, which is a middle-sized binary CNN, the memory reduction was >3.6x.
>
> We hope these answers and the global response have addressed all your queries. Should you have further questions, we are ready to provide additional clarifications promptly.

---

> ### Author Response · Authors · 2023-11-22
>
> Dear Reviewer 2PTe,
>
> With less than 24 hours remaining before the deadline, we kindly request any feedback you might have. We wish to ensure that all your concerns have been adequately addressed in our response letter.
>
> We appreciate your dedication and effort.

---

> ### Author Response · Authors · 2023-11-23
> **Thank you for your dedicated advice**
>
> We have addressed the new concerns the reviewer 832g raised (especially, the use of Adam optimizer). As the deadline is approaching, we kindly request a re-evaluation if all concerns have been resolved. We are truly grateful for the dedicated advice provided in this rebuttal process.

---

### Official Review · Reviewer_832g · 2023-10-31

**Soundness:** 3 good
**Presentation:** 3 good
**Contribution:** 2 fair
**Rating:** 5
**Confidence:** 5

**Summary:**

The submission proposes a way to train BNNS in binary space, i.e., without retaining real-valued weights, by modeling the real-weight distribution and approximating a bit-flip probability for the binary weights. The real-weight distribution is modeled by a Gaussian distribution, from which the hypermasks, the analogous to the learning rate in real-valued training, is sampled and used in the binary space update rule. The method is evaluated by training a neural network with 4 fully-connected layers on Digit, MNIST, CIFAR10, and Tiny-ImageNet datasets. The proposed method shows comparable or better error rates when compared against real-valued training using STE.

**Strengths:**

- The notion of binary-space training is an important future direction for BNNs in my opinion, as training is quite complex and compute-heavy for BNNs at the moment
- The paper reads well, with clear motivations for design choices in the proposed methodology
- Mathematical claims are well-made, with reasonable practical assumptions

**Weaknesses:**

- While I like the overall method of the submission, the experimental section leaves a lot of questions as it is not at the scale of most well-received computer vision or even BNN evaluations.
- Particularly, the error rates on CIFAR10 and Tiny-ImageNet are very high, regardless of wether STE training or the proposed EMP mask training is used. Comparing such underfitted (in my opinion) models does not paint a clear picture.
- The experimental setup is limited to fully-connected layers. While these layers are certainly well utilized, most modern DNNs and BNNs employ either convolutional or transformer layers, which questions the practical applicability of the method.

**Questions:**

- What obstacles do the authors forsee in applying the proposed method to BNNs using convolution layers?

---

> ### Author Response · Authors · 2023-11-20
> **Response to the reviewer 832g**
>
> We thank the reviewer for suggesting an experimental design to better evaluate our method. We have answered each of the reviewer's questions below.
>
> > the experimental section leaves a lot of questions as it is not at the scale of most well-received computer vision or even BNN evaluations.
>
> > Particularly, the error rates on CIFAR10 and Tiny-ImageNet are very high, regardless of whether STE training or the proposed EMP mask training is used. Comparing such underfitted (in my opinion) models does not paint a clear picture.
>
> As the reviewer mentioned, the results for CIFAR-10 and Tiny-ImageNet in Table 3 show underfitting and this is mainly due to the limited expressive power of the 4-layer MLP. To adequately evaluate the training performance of our method (and STE) on the middle-sized datasets, we conducted additional experiments using AdamBNN (MobileNetV1 architecture) and ReActNet (ResNet-18 architecture), and have reported the detail in the global response. In summary of the global response, our method achieved a (top-1) accuracy of 91.94% on CIFAR-10 and 74.64% on CIFAR-100 (we utilized instead of Tiny-ImageNet), which were comparable to STE, demonstrating that our binary-space approach performs sufficiently on the middle-sized datasets.
>
>
> > The experimental setup is limited to fully-connected layers. While these layers are certainly well utilized, most modern DNNs and BNNs employ either convolutional or transformer layers, which questions the practical applicability of the method.
>
> > What obstacles do the authors forsee in applying the proposed method to BNNs using convolution layers?
>
> We agree with the reviewer that many modern BNNs are CNN-based, and the training performance on these architectures is important for the realistic evaluation of training algorithms. Therefore, as described above, we evaluated the binary-space training using the popular ResNet-18 and memory-efficient MobileNetV1, and have reported the results in the global response.
> In the original manuscript, the evaluations of the proposed method were limited to MLP, not because our method is unable to train CNN-based BNNs, but to provide a pure comparison of real- and binary-space training in the most basic structure. In fact, as the additional experimental results (Ex.2 in global response) show, our method has been experimentally demonstrated to achieve comparable performance to STE in training convolutional layers.
>
> We hope these answers and the global response have addressed all your queries. Should you have further questions, we are ready to provide additional clarifications promptly.

---

> > ### Comment · Reviewer_832g · 2023-11-23
> > **Author response**
> >
> > While I appreciate the detailed experimental results, I still have concerns as the numbers for ReActNet (and potentially other architectures) are lower than what was reported in their respective papers. Also, I noticed the authors used SGD when training other BNNs but it is well-known that Adam outperforms SGD, particularly for BNN training, which further questions the experimental results put up by the authors.

---

> ### Author Response · Authors · 2023-11-22
>
> Dear Reviewer 832g,
>
> With less than 24 hours remaining before the deadline, we kindly request any feedback you might have. We wish to ensure that all your concerns have been adequately addressed in our response letter.
>
> We appreciate your dedication and effort.

---

> ### Author Response · Authors · 2023-11-23
>
> Thank you for taking the time for us. We are grateful for the further advices. However, it seems there is a bit of a misunderstanding regarding underfitting, so let us explain in more detail.
>
> In Ex. 1, the lower accuracy of ReActNet are not indicative of our method's lack of training capability, but rather caused by **the limited feature extraction capability of ReActNet**.
> In fact, the following reasons support this:
> 1. In the fine-tuning experiments of Ex. 1, we used a fixed backbone to extract features from CIFAR-10/100, and this backbone employs **officially provided pre-trained weights** trained on ImageNet-1k.
> 2. We conducted a further additional experiment; training the real-valued header by **Linear Probing** with ReActNet backbone, and the performance was 49.15%±0.09 (on CIFAR-10). **The low accuracy of the LP indicates that the features of ReActNet themselves are inseparable**. Note that our method outperformed the LP.
> 3. Our method achieved a comparable performance to baseline (SoTA) real-space training methods, including ReSTE.
> 4. In AdamBNN in Ex. 2, the performance exceeds 91.9% on CIFAR-10 (which cannot be considered underfitting), indicating our method performs well with separable features (, or in full-model fine-tuning).
>
> As already mentioned in the global response, while ReActNet's results indeed seems to be underfitting, we showed the results of ReActNet to demonstrate that **our method achieves comparable performance with other real-space training methods, regardless of the type of backbone used**. The performance in non-underfitting situations has already been demonstrated with AdamBNN, **proving sufficient training capability of our method through experiments**.
>
> Also, regarding the 'respective paper,' no results of **CIFAR-10 fine-tuning experiments** were found in this paper. Therefore, there is **no basis to conclude that these results are lower than those reported in the paper** (in fact, as mentioned above, the performance is higher than linear probing, indicating that the underfitting is a problem with ReActNet's feature extraction).

---

> ### Author Response · Authors · 2023-11-23
> **Additional Experiment using Adam Optimizer**
>
> >Also, I noticed the authors used SGD when training other BNNs but it is well-known that Adam outperforms SGD, particularly for BNN training
>
> Thank you for your valuable advice to improve our experiments. Following the reviewer's suggestion, we have conducted a new experiment using the Adam optimizer.
>
> We conducted the header fine-tuning experiment in Ex.1 **using Adam** as the optimizer. The experimental setup was the same as in Ex.1, and we used AdamBNN trained on ImageNet-1k as the backbone. The results are shown in Table 3. As shown in Table 3, the performance of ReSTE improved by using Adam, but **our binary-space training still achieved the comparable (within the error range) performance to ReSTE**, the SoTA real-space training method.
>
> | Table 3               |  AdamBNN on CIFAR-10 | AdamBNN on CIFAR-100 |
> | :---                  | :---: | :---: |
> | ReSTE w/ SGD          | 85.53±0.34 | 62.40±0.42 |
> | EMP Mask w/ SGD       | 85.63±0.16 | 62.03±0.39 |
> ||||
> | ReSTE w/ Adam         | 85.92±0.39 | 63.36±0.38 |
> | EMP Mask w/ Adam      | 85.80±0.18 | 63.05±0.24 |
>
> Our method can reduce the memory consumption proportional to the number of layers by immediately discarding gradients after propagating them to the layer below; however **it is also possible to combine our binary-space training with Adam**. When using Adam, there is an increase in memory consumption in exchange for performance. Note that, in Ex.2, where the full models are fine-tuned, an optimizer like Adam that retains real gradients of all layers consumes considerable memory footprint, which may not be affordable by low-end devices.
>
> These results and discussion will be added in the appendix.

---

> ### Author Response · Authors · 2023-11-23
> **Thank you for your dedicated advice**
>
> We have also addressed the new concerns the reviewer raised. As the deadline is approaching, we kindly request a re-evaluation if all concerns have been resolved.
> We are truly grateful for the dedicated advice provided in this rebuttal process.
>
> In addition, if there are any additional comments regarding the following statement made by the reviewer, please let us know. We have not been able to confirm this as a fact.
> >lower than what was reported in their respective paper

---

### Official Review · Reviewer_Wfp6 · 2023-11-01

**Soundness:** 2 fair
**Presentation:** 3 good
**Contribution:** 3 good
**Rating:** 6
**Confidence:** 3

**Summary:**

This paper proposes a new optimizer that aims at eliminating the usage of latent floating-point weights when training a binarized neural network. It uses only binary weights, which will significantly reduce the memory footprint. The evaluation of the proposed optimizer is done on MNIST, CIFAR-10, and Tiny-ImageNet datasets using a 4-layer fully connected network. Models optimized by the new optimizer show a close accuracy to those optimized by conventional STE methods using latent weights.

**Strengths:**

1. The effort in the paper indeed aligns with an interesting and important domain of efficiently training a binarized network.

2. The paper makes a good analogy to the SGD algorithm, which eases reading.

**Weaknesses:**

1. The analogy from Figure 1(a) to 1(b) is more like binarizing the gradients in a real-valued space. It is not obvious to the reader that $w_t^{*}$ is the ideal target weights in the binary space since the loss landscape may have been changed a lot after binarization. It needs more discussion or justification.

2. It seems that the proposed update rule cannot be applied to propagating the gradients to 1-bit activations, which means STE is still needed when training a BNN.

3. The experiments need to be done using at least some popular models, such as ResNet.

4. The paper does not take BOP [1] as a baseline. BOP should be a closely related work that aims at not using latent weights for training BNNs.
    * [1] K. Helwegen, Latent Weights Do Not Exist: Rethinking Binarized Neural Network Optimization, NeurIPS’19.

**Questions:**

Questions are included in the weakness section.

---

> ### Author Response · Authors · 2023-11-20
> **Response to the reviewer Wfp6**
>
> We thank the reviewer for the careful reading of our paper and constructive comments in detail. We have addressed each of the reviewer's questions below.
>
> > The analogy from Figure 1(a) to 1(b) is more like binarizing the gradients in a real-valued space.
>
> Let us confirm your question. Do you mean, "In Figure 1 (b), it appears that w is being updated in the real-valued space"? This confusion might have stemmed from the placement of omega in Figure 1(a) and w in Figure 1(b) at identical positions; therefore, we have updated the figure in the revised paper to alter the positioning of the weights.
>
> > It is not obvious to the reader that wt\* is the ideal target weights in the binary space since the loss landscape may have been changed a lot after binarization. It needs more discussion or justification.
>
> The target weight merely represents the direction, toward which a point that reduces the loss is found. Here, we use the word 'merely', since such a point may not be necessarily located in the binary coordinates and w\* might increase the loss. We added a footnote that describes the word choice of 'target weight' so that readers won't experience the same confusion you had.
>
> In addition, previous studies like XNOR-Net and AdamBNN have shown that reflecting changes in the loss landscape of BNNs in the optimizer is crucial for enhancing BNN training performance. Our study mainly focuses on emulating the real-space SGD for BNNs, and the loss landscape problem is not directly addressed; however, the combining these methods with our binary-space training is not difficult. For instance, by using the gradient scaling of the XNOR-Net in our binary-space trainining, we can partially alleviate the change in the loss landscape, and in fact, we used this combination in Ex. 1 in the global response.
>
> >  It seems that the proposed update rule cannot be applied to propagating the gradients to 1-bit activations, which means STE is still needed when training a BNN.
>
> We do use STE in our method, but the real-valued gradients are computed strictly at the points in the binary coordinates. It means we do not hold real weights at all during training.
>
> - For your interest, our proposed random mask relies solely on the direction of the gradient, not its magnitude. As demonstrated in Table 4 of our original paper, this hypermask performs nearly as well as the EMP mask in larger networks. This indicates that our method is effective even when the gradient is binary. Therefore, in future work, if gradients can be computed entirely within the binary space, our random mask could enable complete binary-space training of BNNs, both in terms of weights and gradients. This is a particularly intriguing possibility for the field.
>
> >  The experiments need to be done using at least some popular models, such as ResNet.
>
> Acknowledging the reviewer's feedback, we conducted further experiments with ReActNet, which is based on the well-known ResNet-18 structure, and AdamBNN, which is based on the memory-efficient MobileNetV1 structure. The results of these additional evaluations have been detailed in the global response. Notably, in the fine-tuning experiments (Ex. 2), our method attained a top-1 accuracy of 91.94% on CIFAR-10 and 74.64% on CIFAR-100.
>
> > The paper does not take BOP [1] as a baseline. BOP should be a closely related work that aims at not using latent weights for training BNNs.
>
> We thank the reviewer for suggesting a proper baseline. We have added BOP to the baseline methods in the additional experiment (Ex. 1) shown in the global response and the revised paper.
>
> - Just for the record, BOP by Helwegen et al. is a study that proposed to reinterpret the latent real-valued weight in the BNN as a real-valued accumulator of the gradient. In contrast, our binary-space training eliminates the real-valued accumulator because the l-th gradient can be deleted after computing the (l-1)-th gradient and there is no need to maintain gradients for all layers at the same time. Thus, our binary-space training and BOP are essentially different optimizations.
>
> We hope these answers and the global response have addressed all your queries. Should you have further questions, we are ready to provide additional clarifications promptly.

---

> ### Author Response · Authors · 2023-11-22
>
> Dear Reviewer Wfp6,
>
> With less than 24 hours remaining before the deadline, we kindly request any feedback you might have. We wish to ensure that all your concerns have been adequately addressed in our response letter.
>
> We appreciate your dedication and effort.

---

> ### Author Response · Authors · 2023-11-23
> **Thank you for your dedicated advice**
>
> We have addressed the new concerns the reviewer 832g raised (especially, the use of Adam optimizer). As the deadline is approaching, we kindly request a re-evaluation if all concerns have been resolved. We are truly grateful for the dedicated advice provided in this rebuttal process.

---

### Author Response · Authors · 2023-11-20
**Global response**

We thank the reviewers for their comments and helpful suggestions. We are encouraged that all reviewers appreciated our theoretical contributions. We also noticed that all reviewers were concerned about the experiments in practical aspects. Following the suggestion of Reviewer 2PTe, we have updated our paper with a new evaluation scenario where pre-training is performed on a central server, and fine-tuning is conducted on an edge device. Considering practical applications, we performed two settings:
- (Ex.1) Fine-tuning 2-layer binary MLP header of CNN-based BNNs trained on ImageNet-1k.
- (Ex.2) Fine-tuning full model of CNN-based BNNs trained on ImageNet-1k.

As requested by Reviewers Wfp6 and 832g, we utilized modern/popular CNN-based BNNs as backbones: ReActNet [1] (ResNet-18 architecture) and AdamBNN [2] (MobileNetV1 architecture). We also evaluated the performance of our proposed method when applied to convolutional layers in Ex.2. We also added two baseline methods as requested by Reviewer Wfp6: Binary optimizer (BOP) [3] and Rectified STE [4]. Furthermore, we compared the peak memory usages of the real- and binary-space training in Ex.2, reflecting the point made by Reviewer 2PTe. We conducted these experiments on CIFAR-10/100 instead of Tiny-Imagenet since Tiny-Imagenet is a subset of ImageNet-1k and using the same data distribution for pre-training and fine-tuning is not valid for evaluation. Note that we used the pre-trained weights available on GitHub for the backbones.

First, the results of Ex.1 are summarized in Table 1. The table shows the test accuracies [%]. As can be seen, despite not retaining real-valued weights, our EMP Mask achieved performance comparable to other SOTA real-space training methods for BNNs in the scenario of training binary MLP header.  Each method achieved a test accuracy of approximately 85% on CIFAR-10, and **the problem of underfitting, which was a concern of all reviewers, has been resolved**. Although the performances were lower in ReActNet due to insufficient training of the feature extractor, we included these results to show that **our method can achieve comparable performance regardless of the backbone type**. While our EMP Mask emulates the real-space SGD, the accuracy of the EMP Mask was higher than that of the SGD with naive STE. This is because we applied the gradient scaling of XNOR-Net to the training methods except for the naive STE to fairly compare our method to the SOTA methods which use the gradient scaling.

| Table 1       | ReActNet on CIFAR-10 | AdamBNN on CIFAR-10 | ReActNet on CIFAR-100 | AdamBNN on CIFAR-100 |
| :---          | :---: | :---: | :---: | :---: |
| SGD with Naive STE| 49.09±0.46 | 84.91±0.35 | 26.19±0.14 | 61.92±0.24 |
| BOP           | ***49.93±0.18*** | 85.16±0.14 | ***27.01±0.24*** | ***63.88±0.37*** |
| ReSTE         | 49.47±0.47 | **85.53±0.34** | 26.31±0.26 | **62.40±0.42** |
| EMP Mask (ours) | **49.73±0.29** | ***85.63±0.16*** | **26.41±0.27** | 62.03±0.39 |

Second, the results of Ex.2 are summarized in Table 2. As can be seen, even when fine-tuning the full model, including the convolutional layers, the performance of the binary-space training was comparable to that of the real-space training (within the error bars). These results confirm that **our method also works well to train convolutional layers**. Note that these performances are close to the state-of-the-art [4] (92.55% on CIFAR-10). In addition, **the peak memory usage of our methods was about 27%** of that of the real-space training, and it can be said that our method is efficient for training BNNs on edge devices.

| Table 2       |  | AdamBNN on CIFAR-10| AdamBNN on CIFAR-100|
| :---          | :---: | :---: | :---: |
| SGD with Naive STE| Accuracy [%] | 91.99±0.35 | 74.87±0.48|
| | Peak Memory Usage [MB]  | 151.1 | 151.4 |
| EMP Mask (ours) | Accuracy [%] | 91.92±0.15 | 74.64±0.23|
|| Peak Memory Usage [MB]| 41.6| 42.0|

The changes in the revision are summarized as follows (highlighted in red in the PDF):
- In section 5.1, we have updated Table 3 with these results.
- In the introduction and section 4, we have updated some sentences to clarify that the 33x memory reduction is obtained in the theoretical setting of the number of layers being infinite.
- In Section 3.1, we have added the  justification for the target weight w\*.
- Following the suggestion from Reviewer Wfp6, we have updated Figure 1.

---

> ### Author Response · Authors · 2023-11-20
> **References**
>
> **References**
> [1] Zechun Liu, Zhiqiang Shen, Marios Savvides, and Kwang-Ting Cheng. Reactnet: Towards precise binary neural network with generalized activation functions. In ECCV, 2020.
> [2] Zechun Liu, Zhiqiang Shen, Shichao Li, Koen Helwegen, Dong Huang, and Kwang-Ting Cheng. How do adam and training strategies help bnns optimization? In ICML, 2021.
> [3] Koen Helwegen, James Widdicombe, Lukas Geiger, Zechun Liu, Kwang-Ting Cheng, and Roeland Nusselder. Latent weights do not exist: Rethinking binarized neural network optimization. In NeurIPS, 2019.
> [4] Xiao-Ming Wu, Dian Zheng, Zuhao Liu, and Wei-Shi Zheng. Estimator meets equilibrium perspective: A rectified straight through estimator for binary neural networks training. In ICCV, 2023.

---

### Author Response · Authors · 2023-11-23
**Explanation of underfitiing in the ReActNet in Ex.1**

In Ex.1 in the global response, the accuracies of the ReActNet backbone show somewhat underfitting; however, as already mentioned in the global response, we showed the results of ReActNet to demonstrate that **our method achieves comparable performance with other real-space training methods, regardless of the type of backbone used**. The performance in non-underfitting situations has already been demonstrated with AdamBNN, **proving sufficient training capability of our method through experiments**.

Here, not to confuse the reviewers, we explain that the lower accuracy of ReActNet are not indicative of our method's lack of training capability, but rather caused by **the limited feature extraction capability of ReActNet**.
In fact, the following reasons support this:
1. In the fine-tuning experiments of Ex. 1, we used a fixed backbone to extract features from CIFAR-10/100, and this backbone employs **officially provided pre-trained weights** trained on ImageNet-1k.
2. We conducted a further additional experiment; training the real-valued header by **Linear Probing** with ReActNet backbone, and the performance was 49.15%±0.09 (on CIFAR-10). **The low accuracy of the LP indicates that the features of ReActNet themselves are inseparable**. Note that our method outperformed the LP.
3. Our method achieved a comparable performance to baseline (SoTA) real-space training methods, including ReSTE.
4. In AdamBNN in Ex. 2, the performance exceeds 91.9% on CIFAR-10 (which cannot be considered underfitting), indicating our method performs well with separable features (, or in full-model fine-tuning).

---

### Author Response · Authors · 2023-11-23
**Additional Experiment using Adam Optimizer**

Following the advice from Reviewer 1, we conducted the header fine-tuning experiment in Ex.1 **using Adam** as the optimizer. The experimental setup was the same as in Ex.1, and we used AdamBNN trained on ImageNet-1k as the backbone. The results are shown in Table 3. As shown in Table 3, the performance of ReSTE improved by using Adam, but **our binary-space training still achieved the comparable (within the error range) performance to ReSTE**, the SoTA real-space training method.

| Table 3               |  AdamBNN on CIFAR-10 | AdamBNN on CIFAR-100 |
| :---                  | :---: | :---: |
| ReSTE w/ SGD          | 85.53±0.34 | 62.40±0.42 |
| EMP Mask w/ SGD       | 85.63±0.16 | 62.03±0.39 |
||||
| ReSTE w/ Adam         | 85.92±0.39 | 63.36±0.38 |
| EMP Mask w/ Adam      | 85.80±0.18 | 63.05±0.24 |

Our method can reduce the memory consumption proportional to the number of layers by immediately discarding gradients after propagating them to the layer below; however **it is also possible to combine our binary-space training with Adam**. When using Adam, there is an increase in memory consumption in exchange for performance. Note that, in Ex.2, where the full models are fine-tuned, an optimizer like Adam that retains real gradients of all layers consumes considerable memory footprint, which may not be affordable by low-end devices.

These results and discussion will be added in the appendix.

---

### Meta-Review · Area_Chair_uC2Y · 2023-12-10

**Metareview:**

The paper presents a new way to train Binary Neural Networks (BNNs) using only binary weights, bypassing the need for real-valued weights. This is done by modeling real weights with a Gaussian distribution to create 'hypermasks' for updating binary weights.

**Justification For Why Not Higher Score:**

The authors of the ReActNet paper focus solely on ImageNet-1k results, omitting CIFAR-10 finetuning data. However, this doesn't preclude the possibility of conducting comparative experiments on ImageNet-1k, which would offer a more explicit and fair comparison with ReActNet's ImageNet-1k outcomes. The authors suggest potential success in extending experiments to larger datasets like CIFAR-100. Yet, they don't clarify why their algorithm, which has sophisticated designs, couldn't be scaled to ImageNet-1k. In their experiments, they propose fine-tuning BNNs based on existing ImageNet-1k works, despite their algorithm's emphasis on training BNNs from scratch. This implies that experiments on even larger datasets might be feasible. Regarding the Adam optimizer, the authors acknowledge its superiority over SGD, especially in BNN training. They've added a new table in their response to demonstrate this. However, for a more accurate evaluation, the main experimental section should also be updated to reflect this.

**Justification For Why Not Lower Score:**

N/A

---

### Decision · Program_Chairs · 2024-01-16

Reject